# A pH-driven transition of the cytoplasm from a fluid- to a solid-like state promotes entry into dormancy

Matthias Christoph Munder[1], Daniel Midtvedt[2], Titus Franzmann[1], Elisabeth Nüske[1], Oliver Otto[3], Maik Herbig[3], Elke Ulbricht[3], Paul Müller[3], Anna Taubenberger[3], Shovamayee Maharana[1], Liliana Malinovska[1], Doris Richter[1], Jochen Guck[3], Vasily Zaburdaev[2], Simon Alberti[1]*

[1]Max Planck Institute of Molecular Cell Biology and Genetics, Dresden, Germany; [2]Max Planck Institute for the Physics of Complex Systems, Dresden, Germany; [3]Biotechnology Center, Technische Universität Dresden, Dresden, Germany

**Abstract** Cells can enter into a dormant state when faced with unfavorable conditions. However, how cells enter into and recover from this state is still poorly understood. Here, we study dormancy in different eukaryotic organisms and find it to be associated with a significant decrease in the mobility of organelles and foreign tracer particles. We show that this reduced mobility is caused by an influx of protons and a marked acidification of the cytoplasm, which leads to widespread macromolecular assembly of proteins and triggers a transition of the cytoplasm to a solid-like state with increased mechanical stability. We further demonstrate that this transition is required for cellular survival under conditions of starvation. Our findings have broad implications for understanding alternative physiological states, such as quiescence and dormancy, and create a new view of the cytoplasm as an adaptable fluid that can reversibly transition into a protective solid-like state.

*For correspondence: alberti@mpi-cbg.de

**Competing interests:** The authors declare that no competing interests exist.

## Introduction

The cytoplasm of living cells is highly dynamic and yet exquisitely organized. Maintenance of this state requires a constant input of energy and a metabolism that is far from thermodynamic equilibrium. However, organisms typically live in unpredictable environments and frequently experience conditions that are not optimal for growth and reproduction. Under such conditions, cells must protect themselves by entering into a non-dividing state, generally referred to as dormancy (*Lennon and Jones, 2011*).

Dormancy is defined as a state of reversible cell cycle arrest with reduced metabolic activity and changes in cellular organization (*Lennon and Jones, 2011*). It often involves execution of a developmental program, which culminates in the formation of specialized cell types such as spores, seeds, or cysts. These cell types can endure long periods of nutrient starvation, low temperatures, and even desiccation. Dormancy is also accompanied by extensive changes in cellular architecture, some of which are drastic. For instance, dormant cells have a very low water content, their cytoplasm is densely packed, and they show strongly diminished intracellular dynamics (*Ablett et al., 1999*; *Cowan et al., 2003*; *Dijksterhuis et al., 2007*; *Parry et al., 2014*). However, how cells enter into and recover from such a state is still unresolved.

The current paradigm of cellular biochemistry is based on studies in dilute solutions, often performed with only a handful of proteins. Findings made in such dilute regimes have been extrapolated to the cellular interior. In recent years awareness has been increasing that the cellular

**eLife digest** Most organisms live in unpredictable environments, which can often lead to nutrient shortages and other conditions that limit their ability to grow. To survive in these harsh conditions, many organisms adopt a dormant state in which their metabolism slows down to conserve vital energy. When the environmental conditions improve, the organisms can return to their normal state and continue to grow.

The interior of cells is known as the cytoplasm. It is very crowded and contains many molecules and compartments called organelles that carry out a variety of vital processes. The cytoplasm has long been considered to be fluid-like in nature, but recent evidence suggests that in bacterial cells it can solidify to resemble a soft glass-type material under certain conditions. When cells become dormant they stop dividing and reorganise their cytoplasm in several ways; for example, the water content drops and many essential proteins form storage compartments. However, it was not clear how cells regulate the structure of the cytoplasm to enter into or exit from dormancy.

Now, Munder et al. analyse the changes that occur in the cytoplasm when baker's yeast cells enter a dormant state. The experiments show that when yeast cells are deprived of energy – as happens during dormancy – the cytoplasm becomes more acidic than normal. This limits the ability of molecules and organelles to move around the cytoplasm. Similar results were also seen in other types of fungi and an amoeba. Munder et al. found that this increase in acidity during dormancy causes many proteins to interact with each other and form large clumps or filament structures that result in the cytoplasm becoming stiffer.

A separate study by Joyner et al. found that when yeast cells are starved of sugar, two large molecules are less able to move around the cell interior. Together, the findings of the studies suggest that the interior of cells can undergo a transition from a fluid-like to a more solid-like state to protect the cells from damage when energy is in short supply. The next challenge is to understand the molecular mechanisms that cause the physical properties of the cytoplasm to change under different conditions.

environment is very different from such dilute regimes. One reason for this is that the cytoplasm is densely packed with macromolecules. The overall concentration of macromolecules in the cytoplasm is estimated to be around 200–350 mg/ml (*Ellis, 2001*; *Zimmerman and Trach, 1991*), which amounts to a volume fraction of up to 40%. This dense packing of macromolecules is referred to as macromolecular crowding. How macromolecules remain soluble at such high concentrations inside a cell is unknown, but it presumably involves a fine balance of attractive and repulsive interactions between the different cytoplasmic components.

The highly crowded conditions inside a cell generate an environment with specific physical properties. These properties have traditionally been explored by following the diffusive behavior of tracer particles or organelles. Such particle-tracking approaches have led to the realization that intracellular diffusion is anomalous in cells (*Dix and Verkman, 2008*; *Hall and Hoshino, 2010*; *Luby-Phelps, 2000*; *Tolić-Nørrelykke et al., 2004*). Based on these findings, proposals have been made about the physical nature of the cytoplasm, which has either been described as a hydrogel (*Fels et al., 2009*) or, more recently, as a liquid at the transition to a glass-like state (*Parry et al., 2014*). Because many metabolic reactions and signaling processes take place in the cytoplasm, changes in its physicochemical properties should have far-reaching effects on cellular function and survival.

Recent findings indicate that the organization of the cytoplasm can change considerably, in particular under stress conditions such as starvation. In energy-depleted yeast cells, many proteins and RNAs assemble into microscopically visible structures (*Laporte et al., 2008*; *Narayanaswamy et al., 2009*; *Noree et al., 2010*; *O'Connell et al., 2012*; *Sagot et al., 2006*). These structures may constitute storage depots for proteins and RNAs (*Daignan-Fornier and Sagot, 2011*; *Laporte et al., 2008*; *Sagot et al., 2006*). Indeed, several metabolic enzymes assemble into filamentous structures in response to starvation, and the formation of these filaments leads to enzymatic inactivation (*Petrovska et al., 2014*). Importantly, the enzymes contained in these filaments can be reused,

when cells escape from dormancy. This suggests that cells may regulate the structure of the cytoplasm to enter into and exit from a metabolically inactive state.

In this study, we demonstrate that, in acidic environments, entry into dormancy is triggered by an influx of protons that promotes a transition of the cytoplasm from a fluid- to a solid-like state through widespread assembly of proteins into higher-order structures. We show that this transition arrests the movement of organelles and foreign tracer particles. We provide further evidence that this state of reduced intracellular mobility is required for survival of energy depletion stress. Thus, we propose that organisms have global control mechanisms in place to fine-tune the material properties of the cytoplasm, allowing them to enter into a protective solid-like state, when challenged by extreme environmental conditions.

## Results

### Reduced dynamics of the cytoplasm upon depletion of energy

To investigate how eukaryotic cells enter into a dormant state, we focused on budding yeast, a single-celled organism, which can enter into dormancy upon depletion of energy (*De Virgilio, 2012*; *Gray et al., 2004*; *Neiman, 2011*; *Valcourt et al., 2012*). Because a strong reduction in intracellular dynamics is a hallmark of dormant cells (*Cowan et al., 2003*; *Dijksterhuis et al., 2007*; *Mastro et al., 1984*; *Parry et al., 2014*), we first compared the mobility of different cellular organelles in dividing and dormant yeast cells (*Huh et al., 2003*). Induction of a dormant state was achieved by treating yeast cells with 2-deoxyglucose (2-DG, an inhibitor of glycolysis) and antimycin A (an inhibitor of mitochondrial respiration), a treatment that decreases cellular ATP levels by more than 95% (*Serrano, 1977*). Using single particle tracking (SPT) and mean squared displacement (MSD) analysis we found a striking reduction in intracellular movements upon energy depletion (*Figure 1A and B*).

Tracking of endogenous particles provides only limited information on the material properties of the cytoplasm, because they are often membrane-associated and/or move by active transport. Therefore, foreign tracer particles are better suited as probes of the subcellular environment. However, direct injection methods of foreign particles, as they have been developed for mammalian cells, are not feasible for yeast cells because of their small size. We therefore adopted a technique that relies on a genetically encoded viral capsid protein (µNS), which has been used successfully in bacteria (*Parry et al., 2014*). We could show that GFP-µNS self-assembles into distinct particles in the yeast cytoplasm (*Figure 1—figure supplement 1*, *Video 1*). These particles have bead-like properties with a size-dependent MSD and generalized diffusion coefficient $K$ (*Figure 1—figure supplement 2* and *3*), indicating that they are a valuable tool to study the properties of the yeast cytoplasm. Using GFP-µNS particles and SPT, we found that energy depletion caused a similar reduction in the mobility of these foreign particles (*Figure 1C*). Thus, we conclude that upon energy depletion, the cytoplasm of budding yeast transitions into a state with strongly reduced dynamics.

### A drop in cytosolic pH leads to reduced particle mobility in energy-depleted cells

In higher eukaryotes, ATP-driven processes exert fluctuating forces on the cytoplasm, which lead to random movements of particles and thus cytoplasmic mixing (*Brangwynne et al., 2008*, *2009*; *Guo et al., 2014*). These effects are predominantly driven by motor proteins, which are linked to the cytoskeleton. However, in contrast to mammalian cells, yeast cells have a cell wall, and thus only a rudimentary cytoskeleton, which is primarily based on actin. Importantly, the actin cytoskeleton of yeast disassembles upon starvation (*Sagot et al., 2006*), suggesting that this event may be responsible for the reduced particle mobility by removing tracks for motor-based mixing. To test this, we depolymerized the actin cytoskeleton by adding the drug latrunculin A (LatA) to dividing yeast cells. Indeed, GFP-µNS particle mobility was reduced, but the effect was much less pronounced than under conditions of energy depletion (*Figure 2A*). Next, we treated yeast cells with the drug nocodazole to inhibit microtubule-based motor movements. Again, we only observed marginal effects on particle mobility (*Figure 2B*). This indicates that a lack of active motor-driven movements can only partially explain the reduced particle mobility.

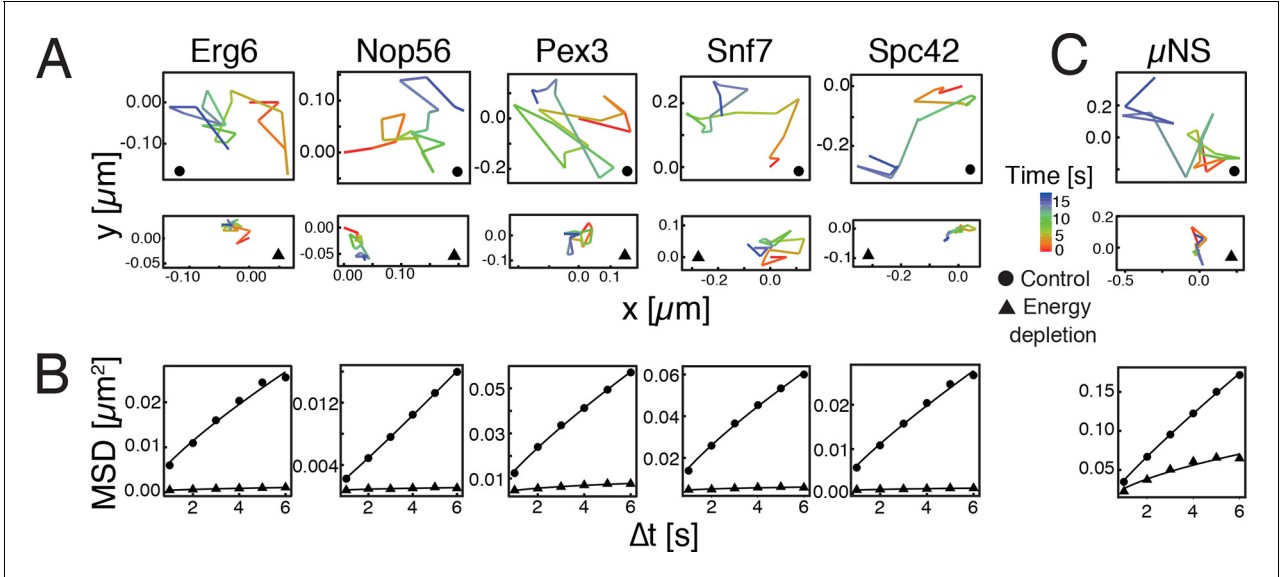

**Figure 1.** Reduced mobility of organelles and foreign tracer particles under energy depletion conditions. (**A**) Representative trajectories of a range of different organelle markers tracked under control conditions (•, cells in log-phase, upper panel) and upon energy depletion (▲, lower panel). Organelle markers are GFP fusion proteins expressed under control of their endogenous promoters (*Huh et al., 2003*). (**B**) Corresponding time- and ensemble-averaged MSD plots for both control (•) and energy depletion (▲) conditions. (**C**) Representative trajectories (upper panel) and corresponding MSD plots (lower panel) of the foreign tracer particle GFP-μNS (see *Figure 1—figure supplement 1–3* for details).

The following figure supplements are available for figure 1:

**Figure supplement 1.** GFP-μNS particles form discrete particles in the yeast cytoplasm (left).

**Figure supplement 2.** GFP-μNS particles show size-dependent MSD over short (left) and long (right) observation times.

**Figure supplement 3.** Same data as in *Figure 1—figure supplement 2*, shown as log-log plots (left).

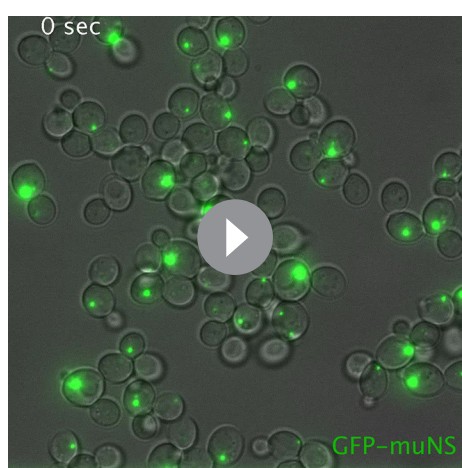

**Video 1.** Time-lapse microscopy of GFP-μNS particles moving in the *S. cerevisiae* cytoplasm. To illustrate how particles explore the yeast cytoplasm over time, the fluorescence channel and the reference bright field channel were merged.

Yeast typically live in acidic environments. The standard laboratory growth media therefore have a pH of around 5.5 (see materials and methods for details). However, the cytosolic pH is kept in the neutral range by proton-translocating ATPases such as Pma1, which use a large amount of energy to continuously pump protons out of the cell, thus preventing cytosolic acidification (*Orij et al., 2011*). In agreement with this, previous studies have reported that energy depletion leads to a drop in cytosolic pH (pHc) (*Dechant et al., 2010*; *Orij et al., 2012*). Indeed, using a ratiometric, pH-sensitive variant of GFP (*Mahon, 2011*) (*Figure 2—figure supplement 1*), we observed a significant pHc decrease from around 7.3 to around 5.8 in yeast cells that were energy-depleted in normal growth medium of pH 5.5 (*Figure 2C*). If this drop in pHc was responsible for the reduced particle mobility, it should be possible to prevent particle immobilization by keeping the pHc in the neutral range. Indeed, when yeast cells were energy-depleted in growth medium of neutral pH, cytosolic

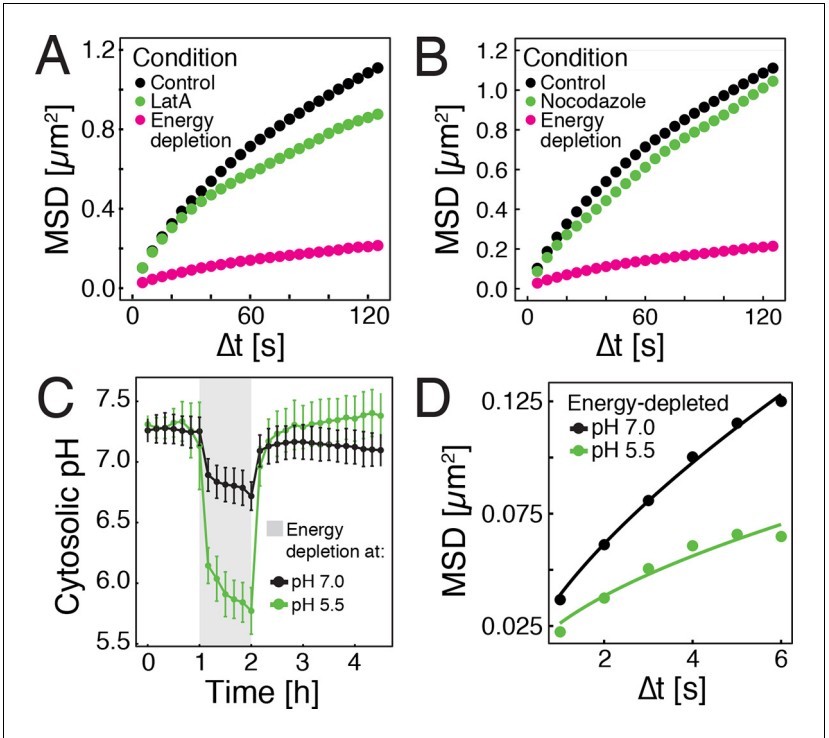

**Figure 2.** Energy depletion causes a drop in cytosolic pH, which may explain reduced particle mobility. (A) MSD of GFP-μNS particles in untreated cells (control), cells treated with 100 μM latrunculin A (LatA) and energy-depleted cells. (B) MSD of GFP-μNS particles in untreated cells (control), cells treated with 15 μg/ml nocodazol and energy-depleted cells (C) The cytosolic pH of yeast cells was measured in response to energy depletion in growth media with two different pHs. (D) MSD of GFP-μNS particles tracked under the conditions shown in C. Cells were energy-depleted in growth medium without glucose containing 20 mM 2-deoxyglucose and 10 μM antimycin A. In panel A and B particles were tracked over a longer time and with lower time resolution (5 s) than in panel D. All MSD plots represent time- and-ensemble averaged MSDs and particles of all sizes were considered.

The following figure supplement is available for figure 2:

**Figure supplement 1.** Yeast cells expressing the pH sensor pHluorin2 in the cytoplasm (left panel) were used to generate a pH calibration curve (right panel), as described in materials and methods and reported previously (*Brett et al., 2005*).

acidification could be prevented (*Figure 2C*) and the reduction in particle mobility was much less pronounced (*Figure 2D*). Thus, we conclude that strong energy depletion leads to a rapid drop in cytosolic pH, which in turn causes reduced particle mobility.

## Reduced particle mobility can be induced by lowering cytosolic pH in the presence of glucose

We next tested whether direct manipulation of the cytosolic pH in the presence of an energy source is sufficient to induce reduced particle mobility. The protonophore DNP rapidly carries protons across the cell membrane and effectively equilibrates the intracellular with the extracellular pH (*Dechant et al., 2010*; *Petrovska et al., 2014*). This allowed us to manipulate the intracellular pH by keeping cells in DNP-containing buffers of different pH (*Figure 3A*, left panel). Cells exposed to DNP-containing buffers generally showed a reduced particle mobility, when compared to cells growing in medium (see *Figure 1C*), most likely because of direct effects of DNP on metabolism. However, the particle mobility was much more strongly reduced at pHc 6 and 5.5 than at pH 7.0 (*Figure 3A*, right panel). To exclude possible secondary effects of DNP, we also used a mild membrane-permeable acid (sorbic acid) to alter pHc (*Orij et al., 2009*) and found that it had a similar effect on cytosolic pH (*Figure 3B*, left panel) and on particle mobility (*Figure 3B*, right panel). These

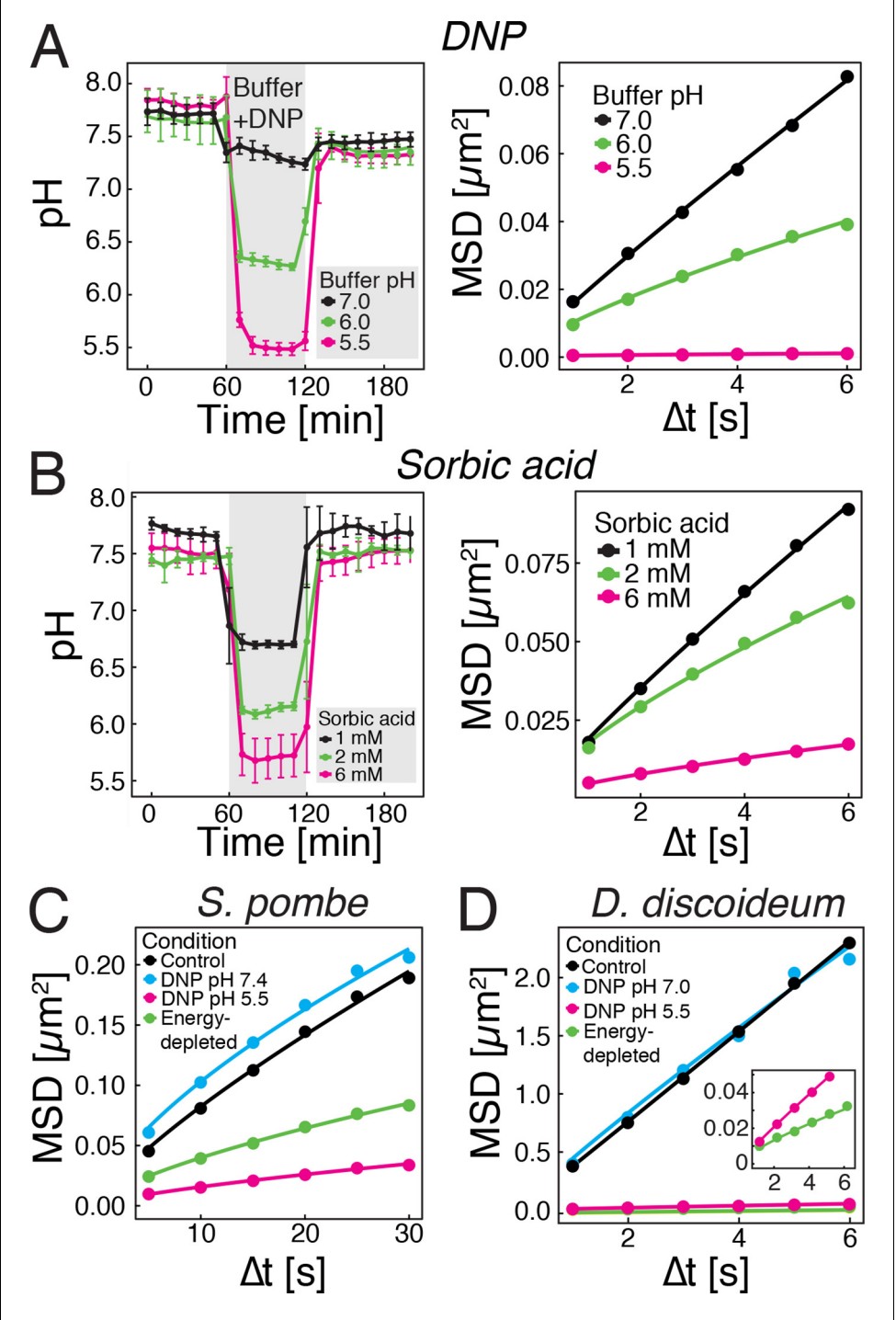

**Figure 3.** Acidification of the cytosol reduces particle mobility. (**A**) The cytosolic pH of yeast cells exposed to phosphate buffers of different pH containing 2 mM 2,4-dinitrophenol (DNP) and 2% glucose was measured over time in a microfluidic flow chamber (left). The MSD of GFP-μNS particles tracked under the same conditions is shown on the right. (**B**) The cytosolic pH of yeast cells exposed to synthetic complete medium (pH ~5.5) containing increasing concentrations of sorbic acid was measured in a microfluidic flow chamber (left). The MSD of GFP-μNS particles tracked under the same conditions is shown on the right. (**C**) MSD of GFP-μNS particles in *S. pombe* cells. Particles were tracked in untreated cells (control), energy-depleted cells and cells treated with phosphate buffers of different pH containing 2 mM DNP. (**D**) MSD of GFP-μNS particles in *D. discoideum* cells. Particles were tracked in untreated cells (control), energy-depleted cells and cells treated with Lo-Flo buffer of different pH

*Figure 3 continued on next page*

*Figure 3 continued*
containing 0.2 mM DNP. Cells were energy-depleted in growth media or buffer, respectively, without glucose containing 2-deoxyglucose and antimycin A. All MSD plots represent time-and-ensemble averaged MSDs and particles of all sizes were considered.

experiments were performed in the presence of glucose as an energy source, suggesting that the pHc change acts downstream of ATP depletion.

Our experimental setup allows for rapid changes of the intracellular proton concentration by almost two orders of magnitude (from 7.4 to 5.5). To test whether such pronounced pH fluctuations affect cell viability, we exposed yeast cells to repeated pHc changes in a microfluidic chamber. Remarkably, pH changes of this magnitude did not affect the viability of yeast (*Video 2*). Moreover, when yeast cells were acidified with DNP, reduced particle mobility manifested within minutes, and it was readily reversed on a similar time scale (*Video 3*). Thus, pH-induced changes are readily reversible and well tolerated by yeast.

Next, we tested whether other eukaryotic organisms undergo similar changes. We focused on another fungus, fission yeast, and a protist, the social amoeba *Dictyostelium discoideum*. As *S. cerevisiae*, both organisms can enter into a dormant state (*Jímenez et al., 1988*; *Sajiki et al., 2009*), form spores upon starvation (*Egel et al., 1994*; *Xu et al., 2004*) and undergo cytosolic pH fluctuations in response to energy depletion (*Gross et al., 1983*; *Karagiannis and Young, 2001*). Consistent with this, both organisms showed reduced GFP-μNS particle mobility in an energy- and pH-dependent manner (*Figure 3C and D*). Thus, we conclude that the pH-induced reduction in particle mobility is not limited to budding yeast, but also extends to other, distantly related organisms.

## Analysis of single particle trajectories indicates a transition of the cytoplasm from a fluid- to a solid-like state

Our findings so far show that the mobility of particles is reduced upon energy depletion and acidification of the cytoplasm. In *Figure 4A*, we show the MSDs and their subdiffusive scaling for different experimental conditions. Particles of all sizes were included in the analysis. We see a dramatic decrease in particle mobility for energy depleted and acidified cells. To gain more insight into the rheological properties of the cytoplasm, which might explain this behavior, we performed a comprehensive analysis of the particle trajectories. From a rheological point of view, the cytoplasm can be considered as an active viscoelastic material (*Guo et al., 2014*; *Mizuno et al., 2007*). The motion of

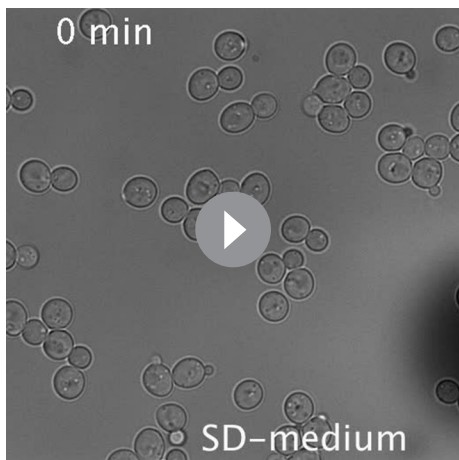

**Video 2.** Brightfield time-lapse microscopy of *S. cerevisiae* cells growing in a microfluidic flow chamber. Cells were exposed to buffers of different pH containing 2 mM DNP as indicated.

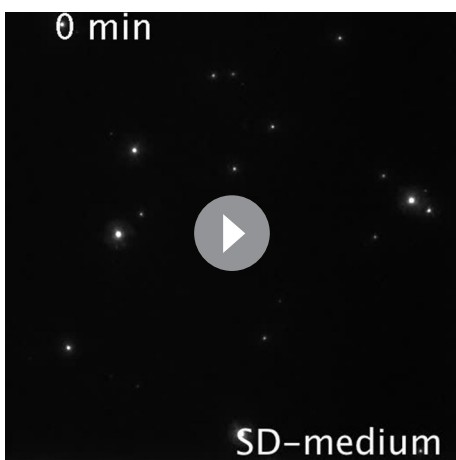

**Video 3.** Fluorescence time-lapse microscopy of GFP-μ NS expressing *S. cerevisiae* cells growing in a microfluidic flow chamber. Cells were repeatedly exposed to buffers of different pH containing 2 mM DNP as indicated.

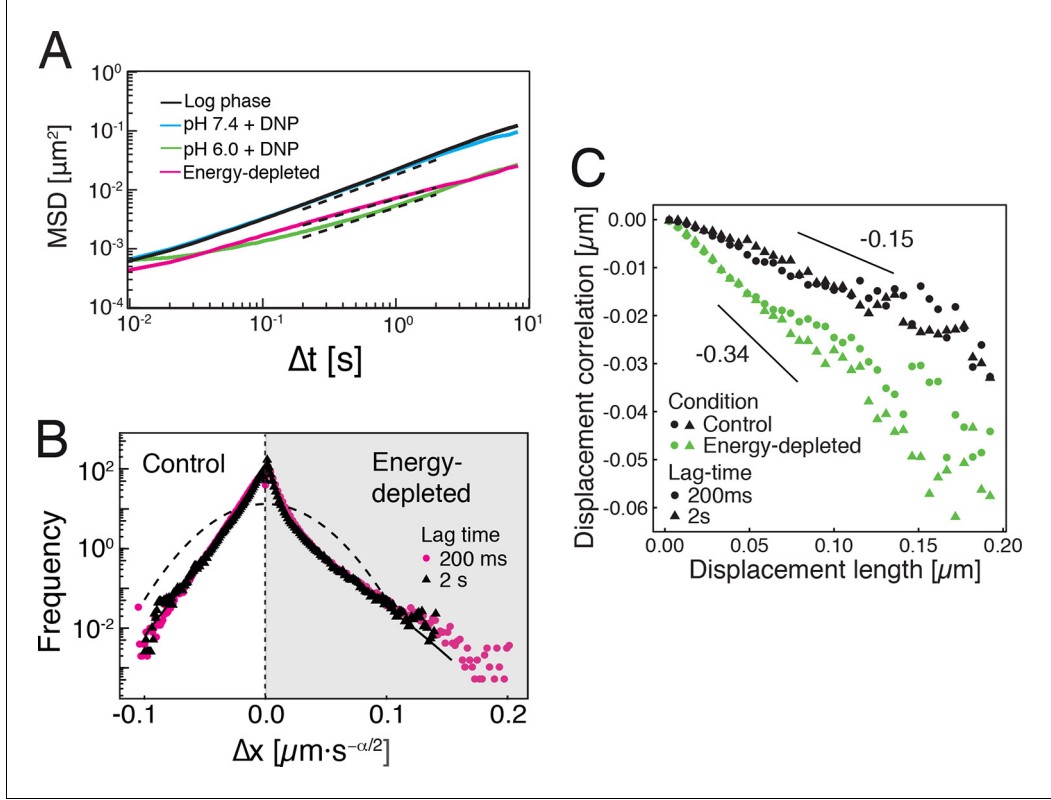

**Figure 4.** Characterization of the acidified and energy-depleted cytoplasm from a particle perspective. (**A**) The time- and ensemble-averaged MSD are shown as a function of lag-time, together with the fitted power-law scalings as dashed lines. We obtained the following power-law exponents $\alpha \approx 0.73$ (DNP treated cells with external pH 6.0), $\alpha \approx 0.84$ (DNP treated cells with external pH 7.4), $\alpha \approx 0.64$ (energy depleted cells) and $\alpha \approx 0.88$ (log phase cells). (**B**) The master curve for the probability density function of particle displacements as a function of rescaled displacement (full lines) for log-phase cells (left side) and energy-depleted cells (right side). Symbols and colors indicate the probability density function extracted from the data at different lag-times after rescaling the displacements. This plot is symmetric for each condition. Plots were split at the dotted line to allow comparison of both datasets. The dashed line corresponds to a Gaussian distribution. (**C**) Correlation of subsequent displacements (defined as $c = \vec{\delta x}' \cdot \frac{\vec{\delta x}}{|\vec{\delta x}|}$, where $\vec{\delta x}$ and $\vec{\delta x}'$ are the displacement vectors at two consecutive time intervals) were calculated from trajectories recorded in control cells and energy-depleted cells for two different lag-times (time used to calculate the displacement) of 200 ms and 2 s. Correlations are plotted as a function of the initial displacement length $|\vec{\delta x}|$. By assuming a linear dependence $c = -b|\vec{\delta x}|$, the slope $b$ quantifies the negative correlations of subsequent particle displacements and its magnitude increases from $b \approx 0.15$ to $b \approx 0.34$ upon energy depletion. The cytosolic pH was adjusted by treating cells with phosphate buffers of pH 5.5, pH 6.0 and pH 7.4, respectively, containing 2 mM DNP and 2% glucose. The cells were energy-depleted in SD-complete medium without glucose containing 20 mM 2-deoxyglucose and 10 µM antimycin A.

The following figure supplements are available for figure 4:

**Figure supplement 1.** Top: The power spectral density (PSD) of particle displacements is shown as a function of frequency $\omega$.

**Figure supplement 2.** The cumulative distribution functions (CDFs) of particle displacements for some representative trajectories in all considered conditions (log-phase, energy-depleted, and pH-adjusted to pH 6.0 and pH 7.4).

**Figure supplement 3.** Left: The probability density functions (PDFs) of particle displacements for all conditions (log-phase, energy-depleted, and pH-adjusted to pH 6.0 and pH 7.4) and at two different lag times after rescaling the displacements both by $t^\alpha$ and the generalized diffusion constant $K_\alpha$.

*Figure 4 continued on next page*

*Figure 4 continued*

**Figure supplement 4.** The directional correlation function, defined as $C(n\tau) = \langle \frac{\vec{\Delta x}(t)}{|\vec{\Delta x}(t)|} \cdot \frac{\vec{\Delta x}(t+n\tau)}{|\vec{\Delta x}(t+n\tau)|} \rangle$, where $\tau$ is the lag-time, $n$ is an integer, $\vec{\Delta x}(t)$ is the change in particle position between time $t$ and $t + \tau$ and $|\vec{\Delta x}(t)|$ denotes the length of the vector $\vec{\Delta x}(t)$, for all conditions (log-phase, energy depleted, and pH adjusted to pH 6.0 and pH 7.4) as a function of $n\tau$ for lag-time $\tau = 200$ ms.

**Figure supplement 5.** Correlations of subsequent displacements (defined as $c = \vec{\delta x}' \cdot \frac{\vec{\delta x}}{|\delta x|}$, where $\vec{\delta x}$ and $\vec{\delta x}'$ are the displacement vectors at two consecutive time intervals) were calculated from trajectories recorded in cells adjusted to pH 7.4 and pH 6.0, respectively, and plotted as a function of the initial displacement length $|\vec{\delta x}|$.

**Figure supplement 6.** Time-averaged (solid lines) and ensemble-averaged (symbols) MSDs of particles in control and energy-depleted cells, respectively (left panel) Time-averaged (solid lines) and ensemble-averaged (symbols) MSDs of particles in cells adjusted to pH 6.0 and pH 7.4 with DNP, respectively (right panel).

---

an inert tracer particle in the cytoplasm results from the balance between stochastic driving forces and opposing material forces. Due to a possible non-thermal origin of stochastic forces in living cells, in general, a combination of passive and active microrheology experiments is required to quantify the material properties of the cytoplasm (*Guo et al., 2014*; *Mizuno et al., 2007*). However, active microrheology experiments in yeast are challenging, because of its small size and stiff outer cell wall. Nonetheless, a detailed analysis of particle trajectories in the passive microrheology approach provides first pointers towards changes in the physical properties of the cytoplasm. One example is the power spectral density (PSD) of particle displacements, which is related to the power spectrum of stochastic forces and the material properties (*Guo et al., 2014*). The PSD can be calculated as a Fourier transform of the MSD and can, for our data in all conditions, be well approximated by a power law dependence $P(\omega) \propto \omega^{-\gamma}$ (see *Figure 4—figure supplement 1*). We find that the exponent $\gamma$ is smaller for acidified and energy-depleted cells. For a viscoelastic material driven by thermal fluctuations, a decrease in $\gamma$ would correspond to a transition from a fluid to a more solid-like state (*Squires and Mason, 2010*). Although we demonstrate that active cytoskeleton-dependent forces do not strongly affect mobility of particles, and we expect no active motion to occur in energy-depleted cells, the thermal nature of driving forces remains an assumption and therefore limits our interpretation of the PSD data. To obtain further insight into particle diffusion, we turn to the analysis of the displacement data on the level of individual trajectories and suggest a statistical model for the observed particle motion.

It has been suggested that the trajectories of tracer particles in cells (*Jeon et al., 2011*; *Tejedor et al., 2010*) and hydrogels (*Stempfle et al., 2014*) can be well described by the model of fractional Brownian motion (fBm). In contrast to ordinary Brownian motion, the displacements in fBm are correlated in time. Positively correlated displacements lead to superdiffusion, whereas negative correlations lead to subdiffusion. As for ordinary Brownian motion, the distribution of displacements of individual particles performing subdiffusive fBm is Gaussian, but the width of the distribution increases with the lag time sub-linearly $\sigma_i^2(t) = 2dK_{\alpha,i}t^\alpha$ (*Hofling and Franosch, 2013*). Here, $\alpha < 1$ is the subdiffusion exponent, $d$ is the dimension of space, and $K_{\alpha,i}$ is the generalized diffusion constant. The subscripts $i$ indicate the diffusivities of individual particle trajectories. We show that our displacement data are consistent with the model of fBm: For sufficiently long individual trajectories, the cumulative distribution functions (CDF) of displacements are well described by CDFs of the corresponding Gaussian distributions (*Figure 4—figure supplement 2*). Also consistent with the model of fBm, the combined (ensemble) distributions of all particle displacements measured for different lag times collapse onto each other after rescaling the displacements as $x/t^{\alpha/2}$ (*Figure 4B*). The value of $\alpha$ is read out from the scaling of the MSD for the corresponding experimental condition (see *Figure 4A*). Remarkably, the shape of this distribution is not Gaussian. However, if we additionally rescale the displacements of each trajectory by its corresponding generalized diffusivity $K_{\alpha,i}$, the combined distributions collapse onto a Gaussian distribution with unit variance (*Figure 4—figure supplement 3*). This collapse shows that the non-Gaussian shape of the combined distribution is a result of the variation in individual particle diffusivities (see materials and methods for details). The

variability of generalized diffusivities could be due to differences in particle sizes, but it could also reflect variations in the properties of the particles' microenvironments. Indeed, we find that even for particles of similar sizes the diffusivities vary strongly for all size groups (see *Figure 1—figure supplement 3*), suggesting that the heterogeneity of the particle microenvironment has a strong impact on particle mobility.

To further test how the microenvironment of the particles changes upon energy depletion and acidification, we analyzed the correlations in the displacements of particles. Negative displacement correlations are the origin of subdiffusive fBm. Indeed, we found that subsequent particle displacements were negatively correlated (see *Figure 4—figure supplement 4* and material and methods for the definition of the correlation function). Interestingly, such negative correlations are a hallmark of particle motion in an elastic environment; particles surrounded by elastic structures tend to be pushed back to their original position. The further the particle is initially displaced, the stronger are the forces pushing it back in the subsequent time interval, which results in stronger negative correlations. Indeed, we find on average that the restoring motion is opposite to the initial step and is linearly proportional to the initial displacement length (*Figure 4C*). In general, the slope of this linear dependence changes from 0 for viscous fluid to -1/2 for an elastic material (*Weeks and Weitz, 2002*). In our data, the magnitude of the slope $b$ increases from $b \approx 0.15$ to $b \approx 0.34$ upon energy depletion and acidification (*Figure 4C* and *Figure 4—figure supplement 5*). This result is consistent with the idea that energy depletion and acidification increase the stiffness of the particles' microenvironments and that the cytoplasm transitions from a fluid-like to a more solid-like state under these conditions.

## Energy-depleted and acidified cells experience a cell volume reduction and display increased mechanical stability

We next tested whether the transition of the cytoplasm to a more solid-like state, as proposed by our particle analysis, also manifests in global changes in the mechanical properties of cells. To experimentally address this question, we mechanically phenotyped budding yeast cells. This required enzymatic removal of the rigid cell wall, which provides mechanical stability to yeast cells, by a process known as spheroplasting. Spheroplasted budding yeast cells were investigated using atomic force microscopy (AFM, [*Radmacher, 2007*]), the standard in cell mechanical characterization, and real-time deformability cytometry (RT-DC, [*Otto et al., 2015*]), a novel microfluidic technique with 100000 times higher throughput. The AFM-based indentation experiments, performed with 10 μm-sized spherical probes to test whole cell mechanics, revealed that acidified cells were about 2.5 times as stiff as control cells (*Figure 5A* and *Figure 5—figure supplement 1*). Importantly, the apparent elastic modulus we measured for spheroplasted cells was three orders of magnitude lower than for yeast cells surrounded by a cell wall (*Figure 5—figure supplement 1*), thus clearly showing that the cell wall had been completely removed. However, cells are usually viscoelastic, and the apparent elastic modulus, as extracted using the Hertz model, reflects their combined elastic and viscous response. To test whether the observed 2.5-fold increase in the apparent elastic modulus of spheroplasts at low pH could also be caused by a strong increase in the viscous resistance to deformation, we extracted the viscosity of the cells from the AFM indentation-retraction curves similar to a recently published method (*Rebelo et al., 2013*; for details see Methods section). We found that the viscosity even decreased from pH 7.4 to pH 6.0 (*Figure 5—figure supplement 2*). Together, the analysis of apparent elastic modulus and viscosity unambiguously demonstrates that the cell body transitions from a compliant, more viscous material to a stiffer, more elastic material at low pH.

The increase in stiffness was independently confirmed by RT-DC measurements, which showed that the mechanical deformability of yeast cells was significantly reduced upon cytosolic acidification (*Figure 5B*). In the RT-DC assay, we also noticed that acidification was associated with differences in cell size, with acidified yeast being smaller (equivalent diameter: 3.503 +/- 0.012 microns; mean +/- SEM; N = 2938) than control cells (equivalent diameter: 3.724 +/- 0.014 microns; mean +/- SEM; N = 2354) (*Figure 5B*). Overall, these whole cell measurements show that acidification changes the mechanical properties of the cells in line with a transition of the cytoplasm to a more elastic, solid-like state, which is accompanied by a reduction in cell volume.

Our findings so far can be explained in two ways: First, cytosolic acidification could lead to a regulatory cell volume decrease, including water loss and increased macromolecular crowding (*Mourão et al., 2014*). Second, acidification could trigger the formation of macromolecular

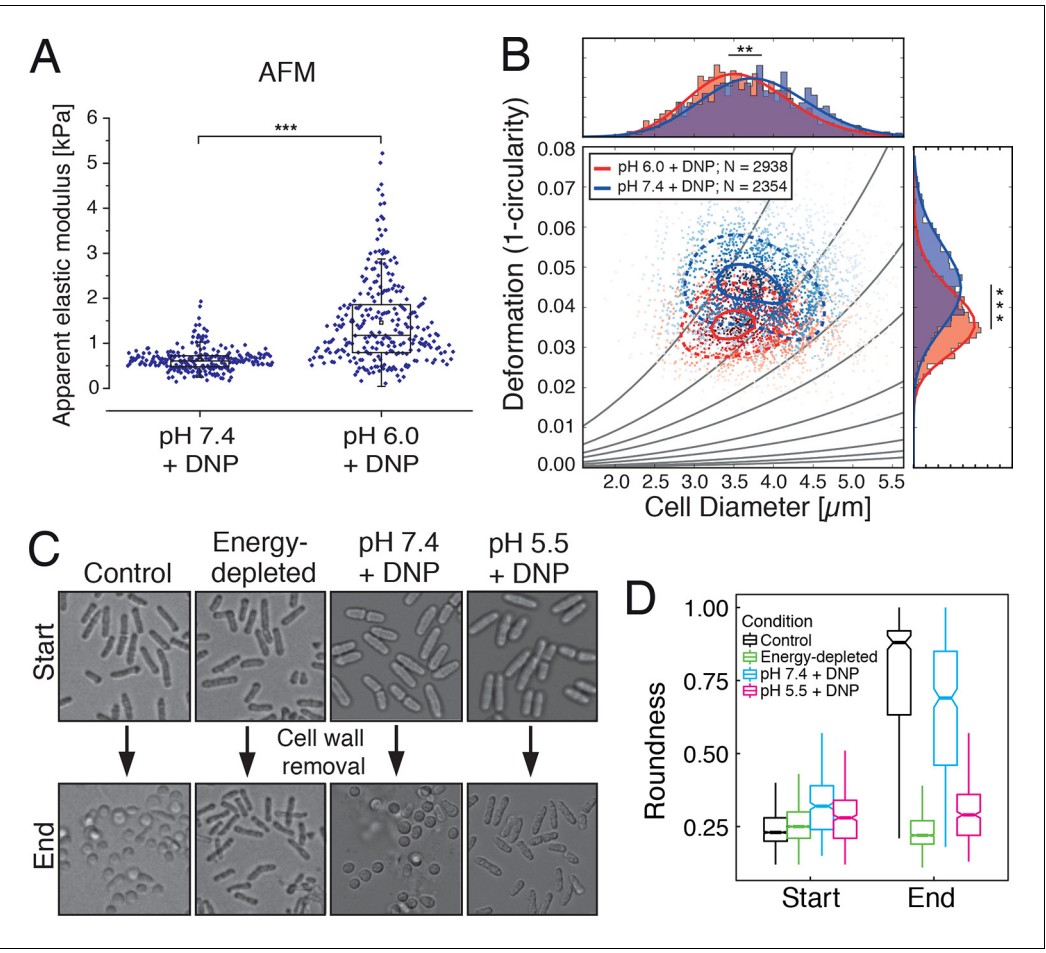

**Figure 5.** Mechanical characterization of acidified and energy-depleted cells. (**A**) The apparent elastic modulus of *S. cerevisiae* spheroplasts (without rigid cell walls) at pH 7.4 ($E = 636 \pm 16$ Pa (mean ± SEM); $N = 249$) and pH 6.0 ($E = 1459 \pm 59$ Pa; $N = 257$) was measured by AFM-based indentation. The cytosolic pH of spheroplasts was adjusted with phosphate buffers of pH 6.0 and pH 7.4, respectively, containing 2 mM DNP, 1% glucose and 1 M sorbitol. (**B**) The same cells as in (**A**) characterized with real-time deformability cytometry (RT-DC). Each measured cell results in a dot in this deformation-cell diameter plot. Also shown are 90% (solid) and 50% (dashed) density lines, and the histograms of size and deformation including Gaussian fits. (**C**) The cell wall of rod-shaped *S. pombe* cells was removed under control, energy depletion, and pH-adjusted conditions. The cytosolic pH of cells was adjusted during spheroplasting with phosphate buffers of pH 5.5 and pH 7.4, respectively, containing 2 mM DNP, 2% glucose, 1 M sorbitol and cell wall-digesting enzymes. Cells were energy-depleted in growth medium without glucose containing 20 mM 2-deoxyglucose and 10 µM antimycin A for 2 hr prior to spheroplasting. Energy depletion was continued during spheroplasting by including 20 mM 2-deoxyglucose and 10 µM antimycin A in the spheroplasting buffer. (**D**) The roundness of more than 160 cells per condition at the start of the experiment and after 3 hr of incubation in the presence of cell wall digesting enzymes (end) was quantified. **p<0.01; ***p<0.001.

The following figure supplements are available for figure 5:

**Figure supplement 1.** (Left) The apparent elastic moduli of *S. cerevisiae* spheroplasts, incubated in phosphate buffers of different pH in the presence and absence of DNP, were determined by AFM-based indentation.

**Figure supplement 2.** The viscosity of the spheroplasted cells, extracted from the AFM-based indentation-retraction curves, decreased from $\eta = 90 \pm 16$ Pa s (mean ± SEM; $N = 31$) at pH 7.4 to $\eta = 70 \pm 14$ Pa s ($N = 23$) at pH 6.0.

**Figure supplement 3.** The volume of pH-adjusted (left panel) and sorbitol-treated (right panel) yeast cells was measured with an imaging-based method (see materials and methods).

*Figure 5 continued on next page*

*Figure 5 continued*

**Figure supplement 4.** Both low pH and sorbitol treatment cause a reduction in particle mobility.

**Figure supplement 5.** The diffusivity of a mCherry-GFP fusion protein was measured with fluorescence recovery after photobleaching (FRAP) in cells exposed to different concentrations of sorbitol (left panel), cells adjusted to different cytosolic pH (middle panel), and in energy-depleted cells (right panel).

**Figure supplement 6.** Energy-depleted *S. pombe* cells retain a rod-like shape in the absence of a cell wall.

**Figure supplement 7.** Cell wall digesting enzymes work equally well in buffers of different pH.

**Figure supplement 8.** Energy-depleted *S. pombe* cells were imaged by time-lapse bright field microscopy after addition of the cell wall removing enzyme mix.

**Figure supplement 9.** Low pH adjusted, rod-shaped *S. pombe* spheroplasts round up when exposed to glucose-containing medium.

---

assemblies, which provide increased mechanical stability to the cytoplasm. In this case, the cell volume reduction could be a result of the exclusion of water from these assemblies (*Cameron et al., 2006*; *Cameron and Fullerton, 2014*; *Fullerton et al., 2006*; *Thirumalai et al., 2012*).

To investigate which scenario might apply, we performed a series of experiments with budding and fission yeast. First, we determined the volume of budding yeast cells using an image-based approach (see materials and methods for details). We found that the cell volume was reduced in a pH-dependent manner. After 30 min at pH 5.5, the cell volume was reduced by ~7% (*Figure 5—figure supplement 3*). Cell volume changes can also be induced by altering the osmotic strength of the growth medium with sorbitol (*Miermont et al., 2013*). Thus, we exposed yeast to different sorbitol concentrations to determine the concentration at which the cell volume was similar to that of acidified yeast. We found that at a sorbitol concentration of 0.8 M the cell volume decrease was of the same magnitude (*Figure 5—figure supplement 3*). As a next step, we compared the particle mobility of osmotically compressed and acidified yeast showing a similar decrease in cell volume. We found that under both conditions particle motion was strongly reduced (*Figure 5—figure supplement 4*). However, in osmotically compressed cells particles of all sizes still performed small movements. These movements could also be detected when yeast cells were exposed to a sorbitol concentration of 1 M, which triggers an even more pronounced cell volume reduction of ~30%. In contrast, particle motion was abolished in acidified yeast (*Figure 5—figure supplement 4*). This suggests that a regulatory cell volume decrease cannot fully explain the reduced particle dynamics of acidified yeast. To further investigate this, we compared the diffusivity of a mCherry-GFP fusion protein (54 kDa) in energy-depleted, acidified and sorbitol-treated cells. The diffusion of mCherry-GFP was not affected in acidified or energy-depleted yeast (*Figure 5—figure supplement 5*), but strongly decreased in cells subjected to high levels of osmotic compression. This indicates the presence of a fluid phase that allows unimpaired diffusion of small macromolecules such as mCherry-GFP in cells exposed to low pH or energy depletion conditions. Thus, cytosolic acidification and osmotic compression seem to induce qualitatively different states of the cytoplasm.

We next analyzed the mechanical stability of fission yeast cells. Fission yeast has an elongated shape, which is supported by the cell wall. However, when the cell wall is removed, fission yeast cells rapidly round up into a spherical shape (*Kelly and Nurse, 2011*; *Sipiczki et al., 1985*). This process requires the cytoplasm to be in a fluid-like state, and it is most likely driven by the osmotic pressure of the cytoplasm and by the passive tendency of the cell to minimize its surface to volume ratio. Remarkably, when we spheroplasted energy-depleted yeast cells, they did not relax into spheres, but maintained their initial rod-like shape (*Figure 5C and D*, *Video 4*). This effect was not due to incomplete removal of the cell wall (*Figure 5—figure supplement 6*, *7* and *8*, *Video 6*). Importantly, for wall-free cells, maintenance of the rod-like shape was pH-dependent and could also be induced by reducing the cytosolic pH with DNP (*Figure 5C and D*, *Video 5*).

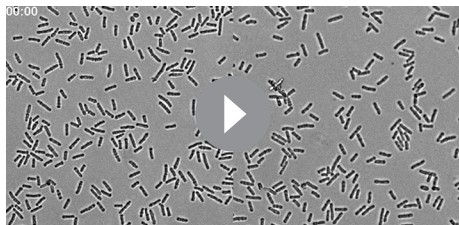

**Video 4.** Brightfield time-lapse microscopy of *S. pombe* cells during cell wall removal. Cells were imaged in EMM5 medium containing glucose (control, left panel) or in EMM5 medium without glucose containing 20 mM 2-desoxyglucose (2DG) and 10 μM antimycin A (right panel). Cell wall removing enzyme mix was added immediately before the recording was started. Medium contained 1 M sorbitol to osmotically stabilize the cells.

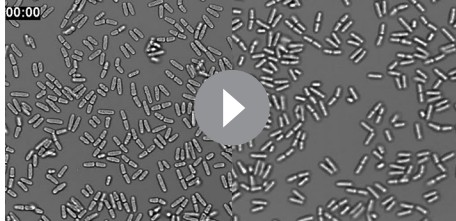

**Video 5.** Brightfield time-lapse microscopy of *S. pombe* cells during cell wall removal. Cells were imaged in phosphate buffer of pH 5.5 (right panel) and pH 7.4 (left) containing 2 mM DNP and 2% glucose. Cell wall removing enzyme mix was added after 30 min of imaging. Buffers contained 1 M sorbitol to osmotically stabilize the cells.

To investigate whether rod-shaped spheroplasts would eventually round up into spheres, we observed them for extended times by time-lapse microscopy. However, the spheroplasts maintained their elongated shape for several hours and did not show signs of rounding up (*Video 4* and *5*). Given this remarkable cellular phenotype, we tested whether the cells are still alive and get softer when energy is provided and the internal pH rebounds to neutral values. Indeed, when acidified yeast cells were re-exposed to medium, the cells quickly became spherical and started to enter the cell cycle (*Figure 5—figure supplement 9*, *Video 7*). Importantly, this rounding up process occurred in the presence of 1 M sorbitol, which was used to osmotically stabilize the cells. Under these conditions, yeast cells experience a substantial reduction in cell volume (*Figure 5—figure supplement 3*), suggesting that an increase in molecular crowding alone does not generate enough mechanical stability to keep the cells in a rod-like shape. Rather, these findings support our idea that cellular stiffening may involve the formation of rigid cytoplasmic structures, which dissolve when energy-depleted yeast re-adjust their cytosolic pH to neutral values. Thus, we conclude that the cytoplasm of energy-depleted cells undergoes a pH-

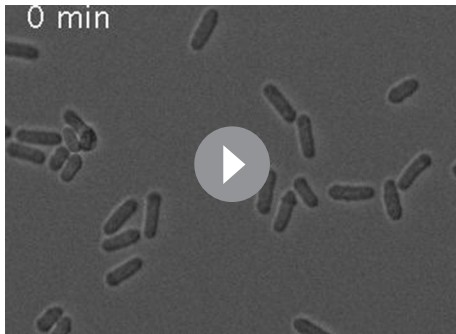

**Video 6.** Brightfield time-lapse microscopy of energy-depleted S. pombe cells during enzymatic cell wall removal. Cells were energy-depleted in EMM5 medium (pH 6.0) without glucose supplemented with 20 mM 2-deoxyglucose and 10 μM antimycin A for 2 hr before imaging. Imaging was then done in cell wall removal buffer (phosphate buffer pH 6.0 containing 20 mM 2-deoxyglucose and 10 μM antimycin A as well as cell wall removal enzymes). Rod-like cells are slipping out of what seems to be a sheath of cell wall material without changing their shape.

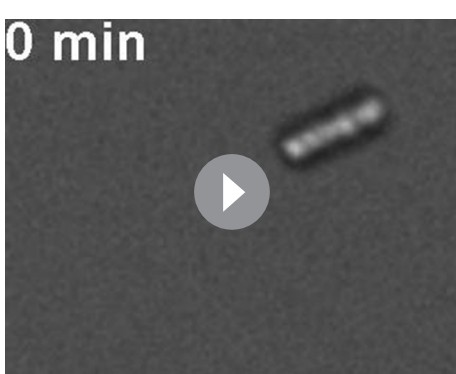

**Video 7.** Brightfield time-lapse microscopy of a *S. pombe* spheroplast. The spheroplast was generated prior to imaging in phosphate buffer of pH 6.0 containing 1 M sorbitol, 2 mM DNP and cell wall digesting enzyme mix and kept in this cell wall removal buffer for 4.5 hr. Directly before the start of the recording, the spheroplast was washed with EMM5 medium containing 1% glucose and 1 M sorbitol. Buffer and medium contained 1 M sorbitol to osmotically stabilize the spheroplast.

dependent transition from a fluid- to a solid-like state, which may be accompanied by the formation of structures that significantly increase the mechanical stability of cells.

## Widespread macromolecular assembly may explain reduced particle mobility and changes in the mechanical properties of cells

Which cytoplasmic structures could be underlying this remarkable change in cellular rigidity? A large number of in vitro studies have shown that the solubility of proteins drops precipitously, when the pH of the solution approaches their isoelectric points (*Tanford and De, 1961*). Under these conditions, proteins interact with each other to form higher-order assemblies, which macroscopically manifest as structures with solid-like properties (*Boye et al., 1996*; *Matsudomi et al., 1991*; *Parker et al., 2005*; *Renard and Lefebvre, 1992*). Thus, we reasoned that the densely packed cytoplasm of yeast cells undergoes a similar transition on a global scale.

To investigate this possibility, we first analyzed the distribution of the isoelectric points of all proteins in the yeast proteome. In agreement with previous work (*Weiller et al., 2004*), we found that the isoelectric points of yeast proteins are largely excluded from the neutral pH range and cluster into two peaks, one in the acidic and one in the basic range (*Figure 6A*). Importantly, the acidic peak overlaps with the pH that cells experience under starvation conditions. This suggests that many proteins have a reduced net charge in energy-depleted cells (*Chan et al., 2006*) and thus become less soluble. This is in agreement with previous results, where it was shown that starvation triggers the assembly of many proteins into higher-order structures (*Narayanaswamy et al., 2009*; *Noree et al., 2010*; *Petrovska et al., 2014*). Importantly, protein complexes remain intact in energy-depleted cells, as shown by the fact that the different proteins in a hetero-complex colocalize in the same structures (*Figure 6—figure supplement 1*). This suggests that the proteins assemble into structures in a native-like state, ensuring that this step is readily reversible. To investigate whether protein assembly and reduced particle mobility are temporally linked, we exposed yeast cells to pH manipulations in a microfluidic chamber and followed assembly and particle movement by fluorescence microscopy. Indeed, we found that these two events coincided (*Video 8*), suggesting a causal relationship.

Given these observations, we reasoned that many proteins might be able to form structures in a pH-dependent manner. Widespread formation of macromolecular protein assemblies could lead to considerable changes in the cellular architecture, and could trigger the formation of a percolated filamentous-colloidal network that would obstruct the movement of particles and provide mechanical stability to the cell. To investigate this possibility, we tested a set of 70 proteins that had previously been shown to assemble into higher-order structures upon starvation (*Narayanaswamy et al., 2009*; *Noree et al., 2010*). We found that the majority of these proteins formed structures upon acidification, whereas such structures were less abundant or absent at neutral pH (*Figure 6B and C*) or in 1 M sorbitol (*Figure 6—figure supplement 2*). Similar structures were observed in dormant yeast spores (*Figure 6B*), which reportedly have a pH in the acidic range (*Aon and Cortassa, 1997*; *Barton et al., 1980*). Thus, we conclude that many proteins assemble into higher-order structures in energy-depleted and acidified cells and that this causes extensive changes in the organization and material properties of the cytoplasm.

## Cytoplasmic acidification promotes survival of energy depletion stress

A hallmark of dormant cells is that they can survive extended periods of energy depletion. We therefore wondered whether pH-induced formation of a solid-like cytoplasm promotes cellular survival. Indeed, when energy-depleted budding yeast cells were kept in the presence of neutral medium to prevent cytoplasmic acidification, they rapidly lost viability (*Figure 7A*). Moreover, when we fixed the pHc in the acidic or neutral range in energy-depleted cells, only acidified *S. cerevisiae* (*Figure 7B*) and *S. pombe* (*Figure 7C*) cells survived. Thus, we conclude that the change in the physical properties of the cytoplasm is protective and that yeast cells use a simple physicochemical signal—the pH of the cytosol—to signal a depletion of energy and to regulate entry into a dormant state.

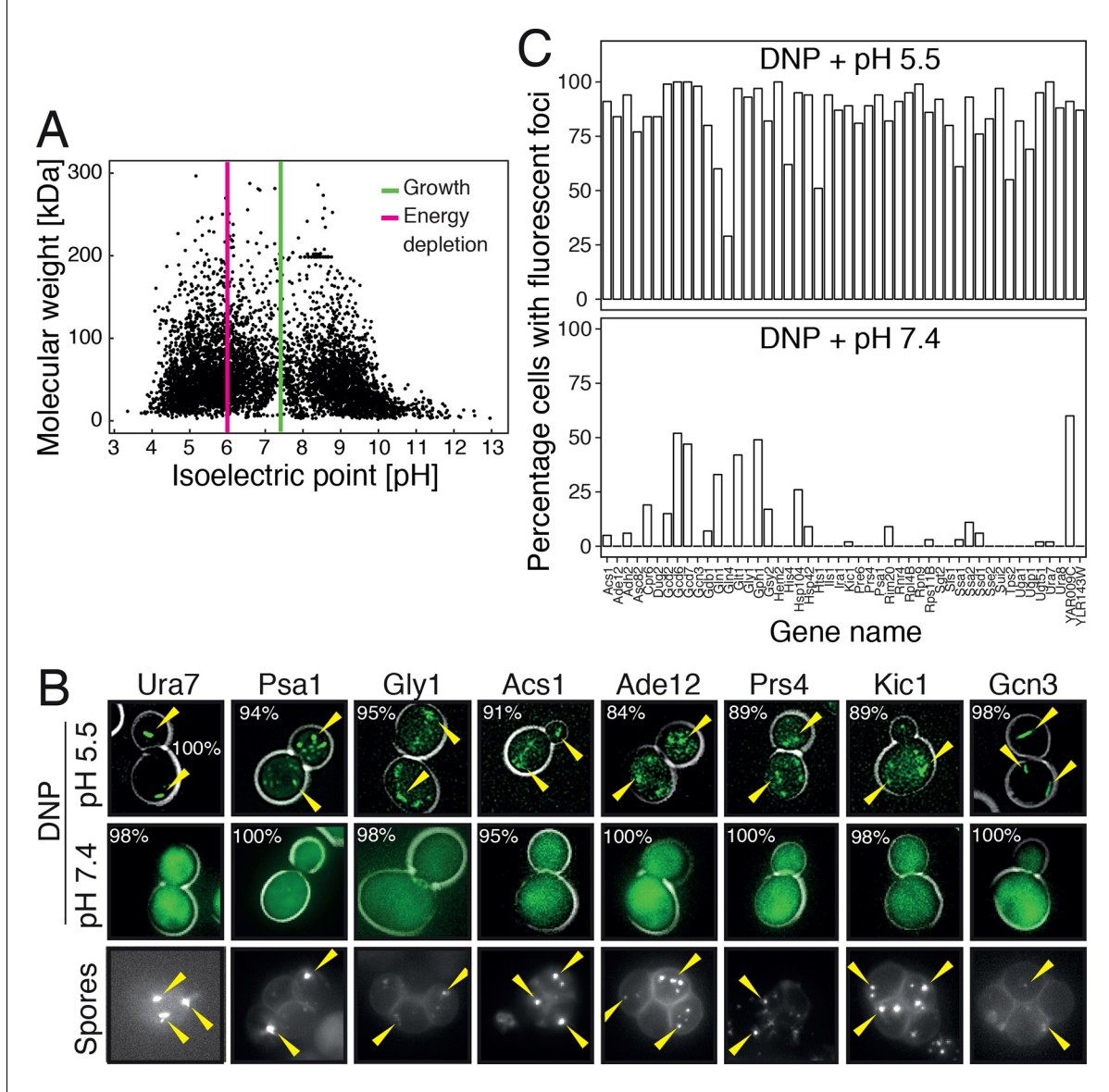

**Figure 6.** Acidification of the cytosol causes widespread assembly of cytoplasmic proteins. (A) The isoelectric points and the molecular weight of all yeast proteins were computed from their primary amino acid sequence and plotted as a virtual 2D gel. The green line indicates optimal growth pH, the red line indicates pH reported for dormant yeast cells. (B) We systematically tested the response of 68 cytoplasmic proteins to a drop in cytosolic pH. Shown are representative images of proteins that responded with assembly formation to low pH. The same proteins also form assemblies in yeast spores. (C) The percentage of cells showing protein assemblies at high versus low pH was quantified. The cytosolic pH was adjusted by treating cells with phosphate buffers of pH 5.5 and pH 7.4, respectively, containing 2 mM DNP and 2% glucose.

The following figure supplements are available for figure 6:

**Figure supplement 1.** Response of 68 cytoplasmic proteins to a drop in cytosolic pH and to treatment with 1 M sorbitol.

**Figure supplement 2.** Different subunits of the hetero-pentameric eIF2B complex colocalize in the same filamentous structures, suggesting that these protein retain their native-like structure when they form higher-order assemblies.

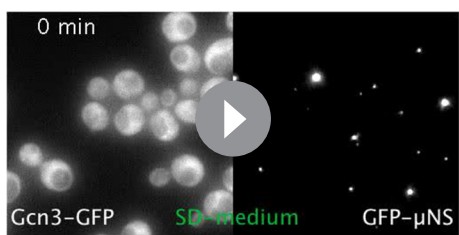

**Video 8.** Fluorescence time-lapse microscopy of Gcn3-GFP (left side) and GFP-μNS (right side) expressing yeast cells growing in a microfluidic flow chamber. Cells were exposed to a phosphate buffer of pH 5.5 containing 2 mM DNP and 2% glucose as indicated.

## Discussion

Many biochemical reactions inside a cell take place in the cytoplasm. Thus, changes in the physical or chemical properties of the cytoplasm will have far-reaching consequences for cellular metabolism and survival. Here, we demonstrate that adaptive changes in the cytosolic pH alter the material properties of the cytoplasm and arrest the diffusion of cellular organelles and foreign tracer particles (see schematic in *Figure 7D*). We further show that pH-controlled macromolecular assembly drives a transition of the cytoplasm to a solid-like state, which provides protection and increased mechanical stability.

Many previous studies have demonstrated a role for energy in controlling intracellular dynamics. In eukaryotic cells, ATP-driven motor proteins carry organelles and other cargo along cytoskeletal tracks to specific subcellular locations, thus regulating the distribution of cytoplasmic components (*Hirokawa et al., 2009*; *Roberts et al., 2013*). Recent findings also indicate that the movements of motor proteins generate random fluctuating forces, which drive diffusive-like non-thermal motion (*Brangwynne et al., 2008*, *2009*; *Guo et al., 2014*). These non-thermal force fluctuations facilitate the mixing of the cytoplasm and thus are important for cellular function. As a consequence, energy depletion in mammalian cells causes a cessation of motor movements and thus a strong impairment

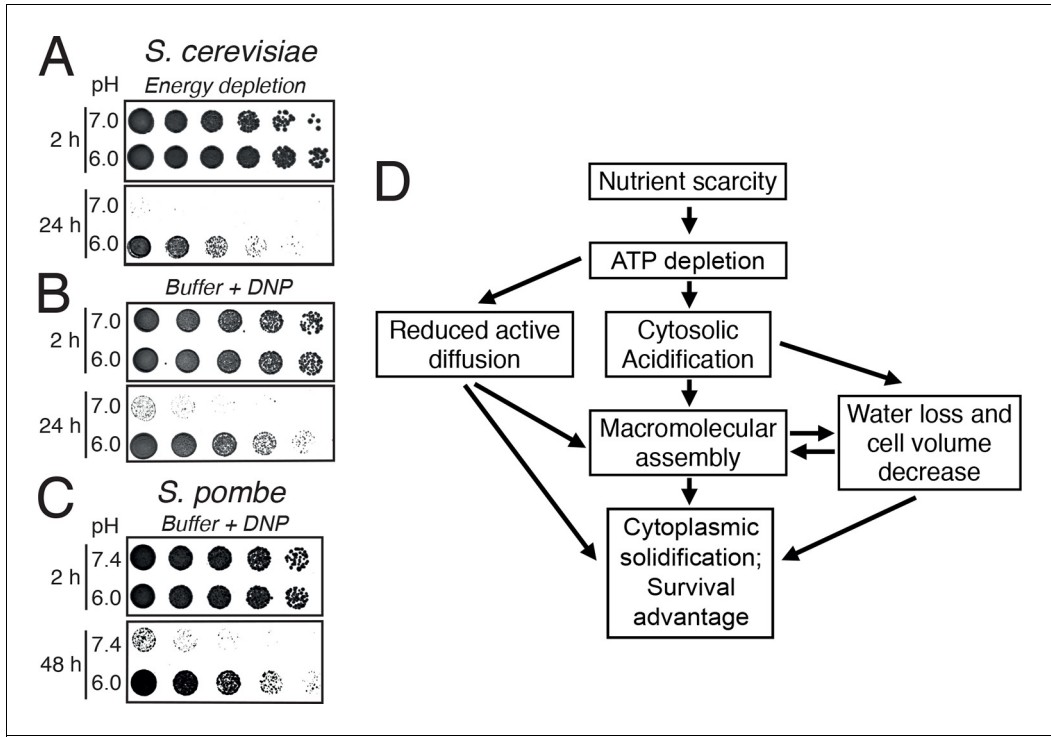

**Figure 7.** Acidification of the cytosol promotes survival under energy depletion conditions. (**A**) Growth assay of *S. cerevisiae* cells that were energy-depleted with 20 mM 2-DG and 10 μM antimycin A in growth medium adjusted to pH 7.0 and 6.0, respectively. (**B**) Similar to A, but cells were incubated in buffers of pH 7.0 or pH 6.0, respectively, containing 2 mM DNP. (**C**) Similar to A, but the experiment was performed with fission yeast. (**D**) A flow chart showing the hypothetical sequence of events promoting entry into a dormant state.

of intracellular motion, with a significant impact on the distribution of macromolecules in the cytoplasm.

The diffusion of a particle in the cytoplasm is the result of two opposing forces: fluctuating forces of thermal or non-thermal origin, which transfer energy onto the particle, thus propelling its motion, and opposing material forces that restrict the movement of a particle and result from interactions of the particle with cytoplasmic components. Studies so far have largely focused on the driving forces of particle motion, and most prominently those that are dependent on ATP (*Brangwynne et al., 2008, 2009; Guo et al., 2014; Parry et al., 2014*). However, we now show that changes in the material properties of the cytoplasm can also significantly affect particle motion. These changes occur in an energy-dependent manner, indicating that energy not only regulates motor-driven diffusive processes, but also has an impact on the organization of the cytoplasm. In the organisms we investigate, these changes are largely independent of the cytoskeleton, but reflect the collective effects of many cytosolic proteins that assemble into higher-order structures.

The chemistry and physics of proteins has been under investigation for many decades. For example, it is a well-known fact that changes in the salt concentration can have a strong impact on protein solubility. One parameter with particularly strong effects on protein solubility is the pH. When the pH of a solution approaches a protein's isoelectric point, the solubility of the protein drops to a minimum (*Tanford and De, 1961*). The reason is that under these conditions proteins attain a low net charge and thus are subject to weaker repulsive interactions. As a consequence, attractive interactions dominate, which—provided that the protein is present at a high enough concentration—can trigger the assembly of proteins into higher-order structures.

The pH-driven assembly of proteins can result in the formation of large structures with solid-like properties (*Boye et al., 1996; Matsudomi et al., 1991; Parker et al., 2005; Renard and Lefebvre, 1992*). Thus, on the macroscopic level, protein assembly can exhibit signs of a phase transition. Although this tendency of proteins to form higher-order structures is well known, this knowledge has not yet been applied to the understanding of living organisms. One reason may be the widespread but misleading assumption that the physicochemical properties of the cytoplasm are invariant. However, in recent years, an increasing number of reports have shown that key physical and chemical parameters of the cytoplasm can fluctuate, especially under conditions of stress. In the case of cytosolic proton concentrations, these fluctuations can span almost two orders of magnitude (our study and *Imai and Ohno, 1995; Orij et al., 2009; Valli et al., 2005*). Given that the cytoplasm is highly crowded with proteins, it is not surprising that pH changes of this scale have strong effects on the physical properties and the dynamics of the cytoplasm.

Previous studies have analyzed the distribution of the isoelectric points in the proteome (*Chan et al., 2006; VanBogelen et al., 1999; Weiller et al., 2004*). These studies found that only few proteins have isoelectric points in the neutral pH range and that most proteins only become electrically neutral when the pH shifts to the acidic or basic range. Importantly, proteins become more insoluble when their net charge decreases. Thus, to maintain maximum proteome solubility, cells have to keep the cytosolic pH in the neutral range. However, when the pH of the cytosol becomes more acidic, as during starvation, a large fraction of the proteome will become less soluble. Consistent with this, we could previously show that cytosolic acidification triggers the assembly of a group of metabolic enzymes into higher-order structures and that assembly inactivates their enzymatic activities (*Petrovska et al., 2014*). Here, we provide evidence that many more proteins assemble into microscopically visible structures upon acidification (*Figure 6B and C*), and we propose that assembly of these and probably many other proteins promotes a transition of the cytoplasm to a more solid-like state. Importantly, pH-dependent assembly of proteins does not seem to go along with protein denaturation, as the proteins in these assemblies retain their native structure (*Petrovska et al., 2014*) and oligomeric states (*Figure 6—figure supplement 2*). This ensures that assembly formation can readily be reversed and that the cytoplasm can rapidly recover from pH-induced alterations, thus allowing swift reentry into the cell cycle. Most importantly, protein assembly and the solid-like state of the cytoplasm protect cells from the adverse effects of energy depletion stress (*Figure 7*). We do not yet know why these processes are protective, but we favor a combination of different explanations, such as energy conservation (*Bernstein et al., 2006*), regulation of metabolism (*Petrovska et al., 2014*), and potentially protection of macromolecules from damage.

One of the hallmarks of dormant cells is a loss of water. Although we do not determine the water content of energy-depleted and acidified cells, we find that acidification goes along with a significant reduction of the cellular volume, which is consistent with a loss of water. Previous studies have shown that protein assembly leads to the exclusion of water (*Cameron et al., 2006*; *Cameron and Fullerton, 2014*; *Fullerton et al., 2006*; *Thirumalai et al., 2012*). The released water becomes osmotically available and can be lost to the surrounding environment, inducing a compaction of the cytoplasm and a change in cell volume. Thus, we propose that the observed cellular shrinkage is to a large degree caused by the formation of cytoplasmic structures and a subsequent release of water.

Additional evidence for this scenario comes from a study that characterized the cytoplasm as a material with distinctive gel-like properties (*Fels et al., 2009*). The authors of this study found that the cytoplasm of mammalian cells behaves like a hydrogel, which can swell and shrink depending on its water content. Importantly, changes in the cytosolic pH could modulate swelling and shrinking (*Fels et al., 2009*). This suggests that the cytoplasm with its many macromolecular components is inherently pH sensitive, a property, which may have been exploited repeatedly during evolution as a strategy for adaptation or survival. In fact, the germination of spores goes along with a drastic increase in water content (*Cowan et al., 2003*; *Dijksterhuis et al., 2007*) and spores consistently have a pH in the acidic range (*Aon and Cortassa, 1997*; *Barton et al., 1980*; *Busa and Crowe, 1983*; *Setlow and Setlow, 1980*; *van Beilen and Brul, 2013*). However, what is still unclear is how water is released from forming spores and re-enters into spores upon germination. Given our findings, we propose that the dehydration/rehydration cycle of spores is at least partially driven by changes in the cytosolic pH. A regulatory cell volume decrease with increased macromolecular crowding may also contribute to the water loss in dormant cells (*Mourão et al., 2014*). Dissection of this important problem will require the use of sophisticated biophysical, biochemical, and genetic approaches.

We show that the cytoplasm of energy-depleted cells transitions from a fluid- to a solid-like state. This transition was evident for the cytoplasm (as determined by particle tracking) and on the level of the entire cell (as determined by cellular deformability assays). This is, to our knowledge, the first viscous and elastic characterization of *S. cerevisiae* spheroplasts by AFM-indentation. The elastic modulus (on the order of 1 kPa) is several orders of magnitude lower than what has been reported for intact yeast cells surrounded by a rigid cell wall (about 500 kPa; [*Pillet et al., 2014*]). A similar difference in stiffness between spheroplasts and intact cells with a rigid cell wall has previously been found in *E. coli* (*Sullivan et al., 2007*). The increased mechanical stiffness at low pH was independently confirmed by a new microfluidic technique (RT-DC). The transition from a compliant, more viscous cytoplasm to a stiff, elastic cytoplasm in energy-depleted yeast cells is in agreement with a model in which many proteins assemble into a dense network, thus restricting the diffusion of large particles. This network could have the overall physical properties of a glass, as recently proposed for bacteria (*Parry et al., 2014*). Future studies will have to determine the molecular mechanisms and physical causes promoting the formation of a solid-like cytoplasm.

In contrast to mammalian cells, yeast cells are much smaller in size. This may explain why yeast rely more strongly on thermal diffusion for macromolecular dispersal. However, this also means that yeast cells have to alter the material properties of the cytoplasm to restrict diffusion during dormancy. We believe this is achieved by promoting a pH-controlled transition to a solid-like state, which significantly changes the fluidity of the cytoplasm. Acidification of the cellular interior of yeast seems to occur through an influx of protons from the outside, suggesting that this transition is dependent on an acidic environment, which may be generated through the normal metabolic activity of yeast. Thus, we predict that single-celled organisms make extensive use of the pH responsiveness of the cytoplasm in order to protect themselves and regulate their metabolism. However, even multicellular organisms such as marine brine shrimp can enter into a dormant state in a pH-dependent manner (*Busa and Crowe, 1983*). Importantly, in this organism dormancy is induced through protons that are released from intracellular stores (*Covi et al., 2005*), indicating that dependence on outside pH could be a peculiarity of yeast. Moreover, cytosolic pH changes have also been observed when organisms such as yeast and *Dictyostelium* are challenged with other types of stresses, such as heat stress or osmotic stress (*Pintsch et al., 2001*; *Weitzel et al., 1985*; *1987*). Thus, global control over the material properties of the cytoplasm through simple physicochemical signals such as the pH could be a frequently used means to regulate cellular function in fluctuating environments.

# Materials and methods

## Strains and culture conditions

S. cerevisiae was grown at 25°C or 30°C in yeast extract peptone dextrose (YPD), synthetic complete (S-complete) or synthetic dropout (SD) medium. Standard pH of SD media is around pH 5.5. S. pombe was grown in either YE5 or EMM5 (standard pH is 6.0) medium at 30°C. D. discoideum was grown in AX medium (ForMedium, standard pH 6.0-6.5) at 23°C under light. A list of all S. cerevisiae strains used in this study can be found in Supplementary file 1. S. pombe wild type strain L972 was used for spheroplasting and spotting experiments. The same strain was transformed with plasmid pDUAL2HFG-µNS-sfGFP for particle tracking experiments. D. discoideum wild type strain AX2-214 (DictyBase) transformed with plasmid pDM353-µNS-GFP (Veltman et al., 2009) was used for particle tracking experiments.

## Plasmids and cloning

A list of all plasmids used in this study can be found in Supplementary file 2. All cloning was done using the Gateway cloning system (Invitrogen) as described previously (Alberti et al., 2007).

## pH adjustment of cells

The intracellular pH of S. cerevisiae and S. pombe cells was adjusted by incubation in phosphate buffers of different pH in the presence of 2 mM 2,4-dinitrophenol (DNP) as described previously (Dechant et al., 2010; Petrovska et al., 2014). DNP was added to the buffers from a 0.2 M (100x) stock solution in methanol. Alternatively, cytosolic acidification was achieved by incubation in SD medium containing 1, 2, or 6 mM sorbic acid. D. discoideum cells were pH adjusted by treatment with either 4 mM sorbic acid or 0.2 mM DNP in LoFlo medium (pH 5.5). For generation of pH calibration curves, cells were treated with 75 µM monensin, 10 µM nigericin, 10 mM 2-deoxyglucose and 10 mM NaN3 in buffers of pH 5.0, 5.5, 6.0. 6.5, 7.0, 7.5, and 8.0 containing 50 mM MES, 50 mM HEPES, 50 mM KCl, 50 mM NaCl, 0.2 M ammonium acetate as described previously (Brett et al., 2005).

## Energy depletion of cells

S. cerevisiae and S. pombe cells were energy-depleted by incubation in SD medium or EMM medium, respectively, without glucose containing 20 mM 2-deoxyglucose (2-DG, inhibition of glycolysis) and 10 µM antimycin A (inhibition of mitochondrial ATP production). This treatment causes a more than 95% reduction in cellular ATP (Serrano, 1977). D. discoideum cells were energy-depleted with 40 mM 2-DG and 200 µM azide in Soerensen-phosphate buffer (pH 6.0).

## Drug treatments

To test the influence of the actin cytoskeleton on particle mobility, cells were treated with 100 µM latrunculin A (LatA) in SD medium for 30 min prior to imaging. To test the role of the microtubule cytoskeleton, cells were treated with 15 µg/ml nocodazole in SD medium for 1 hr prior to imaging.

## Spheroplasting (cell wall removal)

S. cerevisiae spheroplasts were generated by incubating cells in PBS containing 1 M sorbitol (Sigma), 1% glucose and 0.25 mg/mL Zymolyase 100T (USBiological) at 25°C for at least one hour under shaking. S. pombe spheroplasts were generated, with minor adaptations, as described previously (Kelly and Nurse, 2011). Shortly, cells were incubated in buffers of different pH (depending on experiment) in the presence of 1.2 M sorbitol (Sigma), 0.5 mg/mL Zymolyase 100T (USBiological) and 2.5 mg/mL lysing enzymes from Trichoderma harzianum (Sigma).

## S. pombe spheroplasting assay

Wild type S. pombe L972 cells were grown in liquid EMM medium containing 0.5% glucose at 30°C overnight shaking at 200 rpm, diluted and re-grown to mid-log phase the next day. Cells were harvested by centrifugation, washed twice with medium or buffer containing 1.2 M sorbitol and applied to a 4-chamber glass-bottom dish (Greiner BIO-ONE) coated with concanavalin-A. For energy depletion experiments, cells were energy-depleted as described above prior to loading to the dish.

Unbound cells were washed off with EMM medium or phosphate buffers containing 1.2 M sorbitol. Bound cells were covered with 400 µl of phosphate buffer of different pH containing 1.2 M sorbitol and either 2 mM DNP (pH experiment) or 20 mM 2-DG and 10 µM antimycin A (energy depletion experiment). Cells were imaged for five frames before addition of 40 µl cell wall digesting enzymes (final concentrations: 0.5 mg/ml Zymolyase 100T, 2.5 mg/ml lysing enzymes from *Trichoderma harzianum*). Spheroplasting and rounding up of cells was followed by time-lapse bright-field microscopy with a DeltaVision (Applied Precision) microscope (Olympus IX70 stand, Olympus UPlanSApo 20x objective, CoolSnap HQ2 camera).

## Yeast growth assays

*S. cerevisiae* wild type strain W303, or *S. pombe* wild type strain L972 were grown overnight, diluted to $OD_{600}$ ~ 0.1 the next morning and regrown to $OD_{600}$ ~0.5. Cells were harvested and resuspended in either phosphate buffers of pH 6.0 or pH 7.0, respectively, containing 2 mM DNP (*S. cerevisiae* and *S. pombe*) or in S-medium without glucose containing 20 mM 2-DG and 10 µM antimycin A (*S. cerevisiae*). Cells were then incubated under shaking at 25°C. Samples were taken after 2, 24 and 48 hr, cells were washed once with $H_2O$ and subsequently spotted on YPD as five-fold serial dilutions.

## Ratiometric pH measurements

For cytosolic pH measurements a codon-optimized version of the ratiometric fluorescent protein pHluorin2 (*Mahon, 2011*) was integrated into the W303 ADE+ genome at the trp locus. The protein was expressed under control of a GPD promoter. A pH calibration curve was obtained as described previously (*Brett et al., 2005*), except that we used a microscopy-based fluorescence readout. Briefly, cells were incubated in buffers of different pH-containing proton carriers (75 µM monensin, 10 µM nigericin) and inhibitors (10 mM 2-deoxyglucose) to rapidly deplete cells of energy and allow for complete equilibration of the intracellular and extracellular pH. Cells were immobilized in 4-chamber dishes (Greiner BIO-ONE) with concanavalin A and imaged using DAPI/FITC (Excitation: DAPI; Emission: FITC) and FITC/FITC (Excitation and emission: FITC) filter sets on a DeltaVision (Applied Precision) microscope (Olympus IX70 stand, Osram Mercury short arc HBO light source, 100x Olympus UPlanSApo objective, CoolSnap $HQ^2$ camera). Six different Z-stacks each with 6 planes (Z-resolution 0.5 µm) were recorded for each pH condition. Imaging settings were: 10% excitation intensity, 0.1 s exposure time, 512x512 pixels, 2x2 binning. After background subtraction, the mean DAPI/FITC to FITC/FITC ratio was calculated from the intensity readouts of both channels and plotted against pH to obtain a calibration curve. Subsequent pH measurements were calculated from a fourth degree polynomial fit to the calibration curve. Time series of pH measurements were obtained using identical imaging settings and a CellASIC (Millipore) microfluidics flow setup combined with CellASIC ONIX Y04C microfluidic plates.

## Particle tracking experiments

For particle tracking experiments, samples were prepared in 4-chamber glass-bottom dishes (Greiner BIO-ONE). Dishes were coated with concanavalin A coating solution for at least 30 min. Subsequently the coating solution was removed and the glass surface washed with $H_2O$ twice before adding 1 ml of a log phase yeast culture ($OD_{600}$ = 0.5). Cells were allowed to settle onto the glass surface for 10 min. The supernatant was then removed and cells sticking to the surface were washed with appropriate medium or buffer twice. This normally results in a single layer of yeast cells that stick tightly to the glass surface. For control experiments cells were then incubated in 500 µl of S-complete medium for 30 min before imaging. When treated with DNP or sorbic acid cells were incubated in 500 µl of appropriate buffer or medium for 30 min before imaging. For energy depletion experiments, cells were incubated in energy depletion medium for 2 hr before imaging.

Imaging was done on different microscope setups depending on requirements for image acquisition rate and camera chip size. Data with a time resolution of 5 s were recorded on a Deltavision (Applied Precision) microscope (Olympus IX70 stand, Osram Mercury short arc HBO light source, Olympus UPlanSApo 100x oil objective, CoolSnap $HQ^2$ camera, resulting pixel size (x, y) = 65 nm). Z stacks with 10 focal planes were collected at each time point. Imaging settings were: 10% excitation intensity, 0.08 s exposure time, 1024x1024 pixels, total imaging time 10 min. All data with a time

resolution of 1 s were recorded on an Andor spinning disk confocal microscope (Olympus IX81 stand, Andor iXon+ EMCCD camera, resulting pixel size (x, y) = 81 nm). Z-stacks with 16 focal planes were collected at each time point. Imaging settings were: 40% laser intensity, minimum possible exposure time (~16 ms), 512x512 pixels, total imaging time 20 s. Data with 10 millisecond time resolution was recorded on an Andor Spinning disk setup (Olympus IX71 stand, Olympus UPlanSApo 60x silicon oil objective, resulting pixel size (x, y) = 108 nm). Imaging was done in a single focal plane of 764x1190 pixels, which allowed us to track a reasonable number of particles at this frame rate in a single experiment. Imaging settings were: 15% laser intensity, 10 milliseconds exposure time, total imaging time of 10 s.

If recorded, Z stacks were sum-projected using the Fiji image-processing package (*Schindelin et al., 2012*). All particle tracking was done with the MosaicSuite particle tracker (*Sbalzarini and Koumoutsakos, 2005*) a Fiji plugin freely available from http://mosaic.mpi-cbg.de. The following settings were used for tracking data with 5 s time resolution: particle radius: 7, cutoff: 0, percent: variable, link range: 3, displacement: 20. Data with 1 s time resolution was tracked with: particle radius: 8, cutoff: 0, percent: variable, link range: 1, displacement: 20. Data recorded with 10 ms time resolution was tracked with: particle radius: 8, cutoff: 0, percent: variable, link range: 1, displacement: 5. The MosaicSuite particle tracker also measures particle intensities (m0) during tracking. The mean intensity was computed from m0 for each trajectory and used as a proxy for particle size. Particle trajectories were binned into three roughly equally populated size bins (small, medium, large) to illustrate the dependence of the MSD on particle intensity. Computations and plotting were either done in R, making use of the plyr, reshape and ggplot2 (*Wickham, 2009*) packages or in MATLAB.

## Microscopy of protein assemblies

We imaged a list of 73 strains (see *Supplementary file 1*) from the yeast GFP collection (*Huh et al., 2003*) under conditions of high and low intracellular pH. Samples were prepared in 4-chamber glass bottom dishes as described for the single particle tracking experiments. Cells were incubated in phosphate buffers of pH 5.5 or pH 7.4, respectively, containing 2 mM DNP and 2% glucose for exactly 30 min before imaging. Imaging was done on a DeltaVision (Applied Precision) microscope (Olympus IX70 stand, Osram Mercury short arc HBO light source, Olympus UPlanSApo 100x oil objective, CoolSnap HQ2 camera). Z stacks with 14 focal planes were collected at 6 points for each sample. Imaging settings were: 50% excitation intensity, 0.1 s exposure time, 512x512 pixels, 2x2 binning. Imaging of protein assemblies in yeast spores was done with similar settings.

## Cell volume measurements

GFP-expressing yeast cells were used to determine the volume of cells using an imaging-based approach. Samples were prepared in 4-chamber glass bottom dishes as described for the single particle tracking experiments. In control experiments cells were subsequently incubated in 500 µl of S-complete medium for 30 min before imaging. For pH adjustment cells were incubated in 500 µl of phosphate buffers of pH 5.5, 6.5 or 7.4, respectively, containing 2 mM DNP and 2% glucose for exactly 30 min before imaging. For volume adjustment cells were incubated in S-complete medium containing 0.6, 0.8, 1 or 2 M sorbitol for exactly 30 min before imaging. Cells were imaged on an Andor spinning disk confocal microscope (Olympus IX81 stand, Olympus UPlanSApo 100x oil objective, Andor iXon+ EMCCD camera, resulting pixel size (x, y) = 81 nm). Z-stacks were obtained with z=210 nm resolution. Z stacks were projected to obtain 2D maximum intensity projections, which were then processed further for image segmentation and object detection. For image segmentation, objects smaller than 10 pixels were considered to be noise and removed, and a structural filter of ellipsoid shape was applied to detect the foreground of the cells. The background was identified by computing the distance transform matrix of the foreground. Using the watershed transform matrix, the background markers were turned into regional minima, and the foreground image was segmented to obtain individual cells. The mask for the individual cells was then used to select the corresponding Z stack, and the pixels above the stack threshold were considered as resulting from the GFP signal of the cell. Finally, the empty vacuolar regions were filled, and the resultant image was counted for total number of pixels above threshold to compute the total cellular volume. Image

processing and analysis was done in MATLAB. To obtain an accurate measurement of the cell volume, budding and overlapping cells were not quantified.

## FRAP measurements

The mobility of a mCherry-GFP fusion protein was measured using fluorescence recovery after photobleaching (FRAP). Prior to imaging, yeast cells were either pH adjusted in phosphate buffers of pH 5.5, 6.0 or 7.4, respectively, containing 2 mM DNP and 2% glucose, or treated with SD-medium containing 0.8, 1.0, 1.5 or 2.0 M sorbitol, respectively, or energy-depleted in SD-medium without glucose containing 20 mM 2-DG and 10 µM antimycin A. Cells were then immobilized on a cover slip with concanavalin A coating solution and imaged on an Andor spinning disc microscope (Nicon eclipse Ti stand, Nikon Plan Apo TIRF 100x oil objective, Andor iXon+ camera, resulting pixel size 70 nm) equipped with a FRAPPA unit (Andor). A single pixel region of interest was bleached with a 405 nm laser pulse (1 repeat, 40% intensity, dwell time 60 ms). Recovery from photobleaching was then recorded in a single focal plane with a time resolution of 5.4 ms (EM gain 200, laser intensity of 5%). Image analysis was carried out in FIJI.

## Atomic force microscopy

AFM-based indentation experiments were performed using a Nanowizard I (JPK Instruments, Berlin) in combination with the CellHesion module. Tip-less silicone cantilevers (Arrow-TL1, Nanoworld, Switzerland) were equipped with a polystyrene bead of 10 µm diameter (microParticles GmbH, Germany) and calibrated prior to measurements using the thermal noise method. Cell-Tak (Corning, USA) was used for immobilization of spheroplasts (*Gautier et al., 2015*). To determine the stiffness of single spheroplasts (*S. cerevisiae)*, the cantilever was positioned above individual cells and lowered with a speed of 10 µm/s. Force-distance curves were recorded (maximum force 2 nN) and analyzed using the JPK data processing software (JPK instruments): Force-distance data were corrected for the tip-sample separation (*Radmacher, 2007*) and fitted with the Hertz model for a spherical indenter (*Sneddon, 1965*). An effective probe radius was used according to the Hertz model for two spheres. A Poisson ratio of 0.5 was assumed. Experiments were carried out in phosphate buffer (containing 1 M sorbitol and 1% glucose) adjusted to pH 6.0 or pH 7.4 at room temperature both with and without DNP (see *Figure 5* and *Figure 5—figure supplements*). Reporting an *apparent* elastic (Young's) modulus acknowledges the fact that several assumptions of the Hertz model (isotropic, homogeneous, semi-infinite space) are not met; but this still serves well for quantitative comparison of cells in different pH conditions. The Hertz model also assumes an elastic material, but cells are viscoelastic and an observed increase in apparent elastic modulus could also be caused by an increase in viscous resistance to deformation. To directly determine the viscosity $\eta$ of spheroplasted cells from the recorded force-distance data, we adapted a method proposed by (*Rebelo et al., 2013*). Briefly, this method extracts the viscosity $\eta$ by comparing the dissipated energy during the indentation process, $W_{diss}$, to the viscous work, $W_v$, which is modeled taking into account the indenter shape and indenter velocity. $W_{diss}$ corresponds to the area between the approach and retract force-distance curves, $W_{diss} = \int_0^{\delta max}(F^{(app)} - F^{ret})d\delta$, where $\delta$ is the tip-sample-separation, or indentation, and the superscripts *app* and *ret* indicate the forces recorded during approach and retraction, respectively. The viscous work is the integral of the viscous force $F_v$, which is described by $F_v = 2\pi\eta(R - \delta)d\delta/dt$ for a spherical indenter of radius $R$. Force-distance curves were smoothened using a median filter and a multi-exponential fit to compute the time-derivative of the indentation. Finally, the viscosity was calculated as $\eta = \frac{W_{diss}}{2\pi\int_0^{\delta max}(R-\delta)('\delta^{(app)} - '\delta^{(ret)})d\delta}$, where $'\delta = \frac{d\delta}{dt}$. All calculations were implemented in Python.

## Real time deformability cytometry

Real-time deformability cytometry (RT-DC) has recently been introduced as a method for high-throughput cell mechanical characterization (*Otto et al., 2015*). Briefly, the experimental setup consists of an inverted microscope (Zeiss, Axiovert200M), a high-speed video camera (MC1362, Mikrotron) and a syringe pump (NemeSys, Cetoni), which are assembled around a microfluidic chip. The chip is made of polydimethylsiloxane using soft lithography and its geometry is defined by two reservoirs connected by a 300 µm long constriction with a 10 µm x 10 µm squared cross-section. When

a cell suspension is driven through the narrow channel, cells experience a significant hydrodynamic stress and the resulting deformation is captured and analyzed in real-time using the high-speed camera. Deformation, D, is quantified by the circularity c: $D = 1 - c = \frac{2\sqrt{\pi A}}{l}$, where A is the projected surface area and l the perimeter of the cell inside the channel. The more a cell deviates from an ideal circular shape the larger is D. For simplicity, the size measure reported is the diameter of an equivalent circle with area A. A typical experiment requires a cell concentration of $10^6$ cells/ml and a minimal absolute sample volume of 100 μl. Here, *S. cerevisiae* spheroplasts were resuspended in PBS-methylcellulose medium adjusted to different pH, containing 1 M sorbitol and 1% glucose. Cells were drawn into a 1 ml syringe and connected to polymer tubing, which had been cleaned using 70% ethanol and 200 nm sterile-filtered (Sigma Aldrich) deionized water. After connecting tubing to the syringe pump and microfluidic chip a flow was stabilized for 1 min. Here, measurements were carried out at a constant flow rate of 0.012 μl/s. For reference, data are also acquired inside the reservoirs to ensure no deformed cell shape in the absence of mechanical stress (data not shown).

## Mean squared displacement analysis

The ensemble-averaged mean squared displacement (MSD) was calculated as

$$MSD(t) = \frac{1}{N}\sum_{i=1}^{N}\left(x_i(t) - x_i(0)\right)^2 + \left(y_i(t) - y_i(0)\right)^2$$

where $N$ is the number of particles, and $x_i(t)$ and $y_i(t)$ are the coordinates of particle $i$ at time $t$.

The time-and-ensemble averaged MSD, $MSD_\tau$, was computed as

$$MSD_\tau(t) = \frac{1}{N}\sum_{i=1}^{N}\frac{1}{m-t}\sum_{j=0}^{m-t-1}\left(x_i(t+\tau_j) - x_i(\tau_j)\right)^2 + \left(y_i(t+\tau_j) - y_i(\tau_j)\right)^2$$

where $t$ is the frameshift, $N$ is the number of particles and $m$ is the length of the particle trajectory. The maximum frameshift was limited to 1/3 of the full trajectory length. The subdiffusion exponent $\alpha$ was estimated by fitting the time-and-ensemble averaged MSD to a power law between 0.2-2 s. Before fitting, the MSD was noise corrected assuming the positional noise of 11nm estimated from the power spectra. In this way, we obtained $\alpha \approx 0.73$ (DNP treated cells with external pH 6), $\alpha \approx 0.84$ (DNP treated cells with external pH 7.4), $\alpha \approx 0.64$ (energy depleted cells) and $\alpha \approx 0.88$ (log phase cells).

## Analysis of the power spectrum of displacements

The power spectrum of displacements for a one-dimension signal $x(t)$ is given by $\left\langle x(\omega)^2 \right\rangle$, where the angular brackets denote an average over the ensemble, and $x(\omega) = \int_0^\infty x(t)e^{i\omega t}dt$. For the two-dimensional tracking data, the total power spectrum is given by the sum of the power spectra of each component as $\left\langle x(\omega)^2 + y(\omega)^2 \right\rangle$. The integral is approximated using the built-in MATLAB function fft (Fast Fourier Transform). The tracking accuracy is estimated by considering the plateau of the power spectrum. The experimental power spectrum is given by $\left\langle \hat{x}(\omega)^2 + \hat{y}(\omega)^2 \right\rangle = \left\langle x(\omega)^2 + y(\omega)^2 \right\rangle + \left\langle x_0^2 + y_0^2 \right\rangle$, where $\left\langle x_0^2 + y_0^2 \right\rangle$ is the power spectrum of the positional noise. This positional noise will manifest itself as a plateau in the power spectrum at high frequencies. We find that the plateau can be reproduced by generating Gaussian noise with standard deviation of $\sigma \approx 11$ nm. The true power spectrum is obtained by subtracting the power spectrum of this estimated positional noise from the experimentally obtained power spectrum. Thereafter, the low-frequency regime (from 0.1 Hz to 5 Hz) was fitted to a power law in a least square sense. The low frequency regime was used for the fitting as this procedure minimizes potential additional errors from the tracking procedure.

To check whether the deviation of the slope of the PSD from the expected slope from Brownian motion is an artifact of the finite length of the trajectories or a result of correlations in the data, signifying a viscoelastic material state, the displacements of the individual trajectories were randomly reshuffled. This was done by randomly permuting the order of the displacements using the MATLAB function randperm. This way, subsequent displacements in the reshuffled trajectories are

uncorrelated and the corresponding PSD should therefore scale as for Brownian motion. We find that this is indeed the case under all conditions (*Figure 4—figure supplement 1*).

## Fractional Brownian motion as a model for particle diffusion

We find that the time-averaged and ensemble-averaged MSD agrees well under all conditions, implying that the process is stationary (does not age) on the experimental time-scale (*Figure 4—figure supplement 6*). A statistical process that is consistent with our experimental observations (stationarity, subdiffusive scaling of the MSD, and an anomalous power-law scaling of the positional power spectrum (*Weiss, 2013*) is fractional Brownian motion (fBm). For fBm, the probability density function (PDF) of displacements is Gaussian, but the displacements are correlated, $\langle x(t)x(s)\rangle = K_\alpha[t^\alpha + s^\alpha - |t-s|^\alpha]$, where $K_\alpha$ measures the strength of the fBm and may depend on particle size and the local microenvironment. As a consequence of the Gaussian property, the PDF is completely determined by its second moment, proportional to the MSD of the particle. The correlations cause the MSD to increase as a power law, $MSD_\tau(t) = 2dK_\alpha t^\alpha$ where $d$ is the dimension of space, and $0 < \alpha < 1$ is the subdiffusion exponent. From this expression we see that $K_\alpha$ can be referred to as a generalized diffusion constant. In the following discussion, we consider the motion along the two coordinate axes as two independent realizations of the same random process. The motion is consequently analyzed as a one-dimensional process, $d = 1$.

## CDF of particle displacements

For one-dimensional fBm the PDF of displacements of each individual particle is given by $P_i(x,t) = \frac{1}{(4\pi K_{\alpha,i}t^\alpha)^{1/2}}\exp\left(-\frac{x^2}{4K_{\alpha,i}t^\alpha}\right)$. The lengths of the individual trajectories are too short (~1000 time points) to provide a reliable estimate of the PDF on a single-particle level. A statistical measure that can be estimated also for small datasets (*Weiss, 2013*) is the cumulative distribution function (CDF). The CDF $F_i(x,t) = \int_{-\infty}^{x} P_i(x',t)dx'$ of a single particle trajectory is a measure for the probability that the particle makes a displacement not larger than $x$ (note that $x$ has both negative and positive values). To build a CDF $F_i(x,t)$ at a given moment of time ($t = 2$ s) for an individual trajectory we just count the number of displacements which are smaller than $x$ and divide this number by the total number of displacements in a given trajectory. The CDFs are fitted to a Gaussian CDF with zero mean and variance given by the variance of the displacements in the array (see *Figure 4—figure supplement 2*). In addition, for each individual trajectory we can calculate the so-called non-Gaussian parameter $\gamma_i = \frac{\langle \triangle x^4\rangle_i}{3\langle \triangle x^2\rangle_i^2} - 1$, which vanishes for displacements $\triangle x$ with the Gaussian distribution. Indeed we see that for all experimental conditions this parameter is close to zero, see inset in *Figure 4—supplement figure 2*.

## Scaling property and the master curve

By rescaling the displacements by the lag time as $\tilde{x} = x/t^{\alpha/2}$, the PDF of displacements at different times can be collapsed to a single master curve $G_i(\tilde{x}) = \frac{1}{(4\pi K_{\alpha,i})^{1/2}}\exp\left(-\frac{\tilde{x}^2}{4K_{\alpha,i}}\right)$. For an ensemble of $N$ particles performing fBm in a heterogeneous environment, the total master curve is given by $G(\tilde{x}) = \left(\frac{1}{N}\right)\sum_{i=1}^{N} G_i(\tilde{x})$. To obtain the master curve $G(\tilde{x})$ for the fBm process underlying the particle motion, we need to estimate the generalized diffusion constants $K_{\alpha,i}$ of fBm for each trajectory. The strength of the fBm is obtained as $K_{\alpha,i} = \frac{MSD_{\tau,i}(t)}{2t^\alpha}$, for a lag time $t = 2$ s. As we see on *Figure 4B* the master curve perfectly fits the ensemble data for the PDF of particle displacements. Moreover, if we rescale displacements of each trajectory by the corresponding generalized diffusion constant, they should all collapse on a single Gaussian distribution with a unit variance, and this is what we show in *Figure 4—figure supplement 3*). For reference we also provide the unscaled PDFs of displacements for log phase and energy depleted cells for two lag times $t$=0.2 and 2.0 s, see *Figure 4—figure supplement 3*, right panel.

## Displacement correlation

The directional correlation function of the displacements was calculated as $C(n\tau) = \langle \frac{\Delta \vec{x}(t)}{|\Delta \vec{x}(t)|} \cdot \frac{\Delta \vec{x}(t+n\tau)}{|\Delta \vec{x}(t+n\tau)|} \rangle$, where $\tau$ is the lag-time, $n$ is an integer, $\Delta \vec{x}(t)$ is the change in particle position between time $t$ and $t + \tau$ and $|\Delta \vec{x}(t)|$ denotes the length of the vector $\Delta \vec{x}(t)$. Averaging is performed over the ensemble of particles and time $t$.

To quantify the strength of correlations we considered two displacements $\delta \vec{x}$ and $\delta \vec{x}'$ during two consecutive time intervals of length $\tau$ (*Weeks and Weitz, 2002*). We then calculate the projection of the second displacement onto the direction of the preceding one, $c = \delta \vec{x}' \cdot \frac{\delta \vec{x}}{|\delta x|}$. If this value is negative, it indicates that the second displacement tends to move oppositely to the first. For small displacements, we expect this quantity to be a linear function of the initial particle displacement, $c = -b|(\vec{\delta x})|$ . For a viscous material the slope $b$ vanishes, whereas for an elastic material the slope is $b = 1/2$ and is independent of the lag time $\tau$ for which the displacements are calculated. To find out how $c$ depends on the magnitude of the initial displacement, we first extract all pairs of subsequent displacements at a certain lag time. Thereafter the projection is calculated for each pair of displacements. In order to consider the relation between the correlation $c$ and the initial displacement length $|\vec{\delta x'}|$, these lengths were binned in 38 equidistant bins. The correlation was averaged in each bin to obtain the correlation $c$ as a function of displacement length $|\vec{\delta x'}|$, see *Figure 4C* and *Figure 4—figure supplement 5*). The linear scaling of $c$ with $|\vec{\delta x'}|$ and its independence on the lag time also rule out the localization error as a possible (dominating) source of negative correlations in the displacement data.

## Acknowledgements

We thank several members of the MPI-CBG and Christoph Weber from the MPI-PKS for critical reading of the manuscript. We are grateful to Eli Barkai and Daniela Frömberg for helpful discussions. We thank Cammie Lesser for the GFP-μNS plasmid. The light microscopy facility of the MPI-CBG is acknowledged for expert technical assistance. MM was supported by a DIGS-BB doctoral and a springboard-to-postdoc fellowship. SM was supported by a postdoctoral fellowship by the Alexander von Humboldt Foundation. JG was supported by an Alexander von Humboldt Professorship by the Alexander von Humboldt Foundation. We acknowledge founding by the Max Planck Society and the German Research Foundation (DFG, AL 1061/5-1).

## Additional information

### Funding

| Funder | Grant reference number | Author |
| --- | --- | --- |
| Max-Planck-Gesellschaft | Core Funding | Daniel Midtvedt<br>Titus Franzmann<br>Doris Richter<br>Vasily Zaburdaev<br>Simon Alberti |
| Dresden International Graduate School for Biomedicine and Bioengineering | Graduate Student Fellowship | Matthias Christoph Munder |
| Dresden International Graduate School for Biomedicine and Bioengineering | Sprinboard-to-Postdoc-Fellowship | Matthias Christoph Munder |
| Alexander von Humboldt-Stiftung | Alexander von Humboldt Professorship | Oliver Otto<br>Maik Herbig<br>Elke Ulbricht<br>Paul Müller<br>Anna Taubenberger |

| Alexander von Humboldt-Stiftung | Postdoc Fellowship | Shovamayee Maharana |
| --- | --- | --- |
| Deutsche Forschungsgemeinschaft | Reserach Grant, AL 1061/5-1 | Elisabeth Nüske Simon Alberti |

The funders had no role in study design, data collection and interpretation, or the decision to submit the work for publication.

## Author contributions

MCM, Performed the pH measurements and analyzed the data, Imaged the formation of assemblies in yeast cells and spores, Devised an image-based assay to measure cell size, Performed the particle tracking experiments, Performed and analyzed the experiments with spheroplasted fission yeast, Performed and analyzed the particle tracking experiments with Dictyostelium, Conception and design, Acquisition of data, Analysis and interpretation of data, Drafting or revising the article; DM, Analyzed the particle tracking data, Developed a model to describe particle motion, Conception and design, Analysis and interpretation of data; TF, Performed and analyzed the particle tracking experiments with fission yeast, Performed and analyzed the experiments with spheroplasted fission yeast, Conception and design, Acquisition of data, Analysis and interpretation of data; ENske, Imaged the formation of assemblies in yeast cells and spores, Acquisition of data, Analysis and interpretation of data; OO, Performed the RT-DC measurements of spheroplasted yeast, Analyzed RT-DC data, Acquisition of data, Analysis and interpretation of data; MH, Analyzed RT-DC data, Acquisition of data, Analysis and interpretation of data; EU, Conducted the AFM measurements of spheroplasted yeast, Analyzed AFM data, Acquisition of data, Analysis and interpretation of data; PM, AT, Analyzed AFM data, Acquisition of data, Analysis and interpretation of data; SM, Devised an image-based assay to measure cell size, Acquisition of data, Analysis and interpretation of data; LM, Performed and analyzed the particle tracking experiments with Dictyostelium, Acquisition of data, Analysis and interpretation of data; DR, Generated strains and constructs, Acquisition of data, Analysis and interpretation of data; JG, Analyzed AFM data, Analyzed RT-DC data, Conception and design, Analysis and interpretation of data, Drafting or revising the article; VZ, Analyzed the particle tracking data, Developed a model to describe particle motion, Conception and design, Analysis and interpretation of data, Drafting or revising the article; SA, Conception and design, Analysis and interpretation of data, Drafting or revising the article

## Author ORCIDs

Matthias Christoph Munder, http://orcid.org/0000-0003-3594-4725
Jochen Guck, http://orcid.org/0000-0002-1453-6119
Simon Alberti, http://orcid.org/0000-0003-4017-6505

# Additional files

### Supplementary files

- Supplementary file 1. List of strains used in this study.

- Supplementary file 2. List of plasmids used in this study.

- Supplementary file 3. Table with osmolality values of media and buffers.

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
