## [Decision Letter]

[Editors’ note: this article was originally rejected after discussions between the reviewers, but the authors were invited to resubmit after an appeal against the decision.]

Thank you for choosing to send your work entitled "A sol-gel transition of the cytoplasm driven by adaptive intracellular pH changes promotes entry into dormancy" for consideration at *eLife*. Your full submission has been evaluated by Vivek Malhotra (Senior editor) and two peer reviewers, one of whom is a member of our Board of Reviewing Editors, and the decision was reached after discussions between the reviewers. Based on our discussions and the individual reviews below, we regret to inform you that your work will not be considered further for publication in *eLife*.

While both reviewers found the work of interest and importance, they were unconvinced that the data supported the conclusions and felt there were numerous flaws in the experiments and analysis. One reviewer summarizes: "To me, there are two major problems. First, the authors seemed to propose a 'gel' transition originating from an increase in (presumably cross-linked?) protein filaments in the cytoplasm under starved/acidified conditions, but I find the evidence unconvincing. Second, the AFM and cell shape retention results (that the authors interpret as an increase of stiffness of the cytoplasm) could be due to the presence of residual cell wall." In view of these considerations, we do not feel that the manuscript warrants further consideration unless these concerns can be resolved.

*Reviewer #1:*

This study shows that carbon starvation and acidification of the cytoplasm change the physical properties of the cytoplasm of yeast and Dictyostelium cells. Cytoplasmic acidification appears sufficient to reduce the motion of large tracer particles. It also increases survival rate under energy-depleted conditions (i.e., starvation), showcasing the biological significance of the findings. The authors propose that particle confinement under starvation is caused by the cytoplasm transitioning from a solution to a gel-like state. The topic is important and has large implications to our understanding of cellular dynamics and dormancy in eukaryotes. While the results are potentially exciting, there are major concerns with some interpretations and conclusions.

First, the proposed terms 'cytoplasmic freezing' are misleading. If what they observe was analogous to the cytoplasm freezing, all cellular components, including proteins, would reduce their motion as the dynamics of particles of all sizes are affected by temperature. This is not the case here as large particles are affected but not proteins (GFP). The size dependence is inconsistent with a cytoplasmic freezing, and this characterization should be avoided.

The evidence for a gel transition is not convincing. A gel suggests a meshwork. The authors argue that many proteins assemble into higher order structures under starvation, citing Noree et al. (2010), Narayanaswamy et al. (2009) and their own previous work. The Noree et al. (2010) paper talks about a wide genetic screen identifying metabolic proteins that form 4 distinct filaments, with some being induced by energy depletion, but others being unaffected or even reduced upon energy depletion. The Narayanaswamy et al. (2009) paper and Figure 6 here show mostly puncta forming, but puncta (compact aggregates) do not form gels. I agree that a few filaments form under energy-depleted/acidified conditions. But it is unclear that there are more of them under these conditions than under the normal situation, especially given the loss of cytoskeletal filaments under energy-depletion/acidification? To form a gel and trap probes, the cell would need a pervasive meshwork (cross-linked filaments), not just proteins that form higher-order structures.

Figure 4: The method used to show that elasticity of the cytoplasm is increased under acidification is based on a model that explicitly assumes elasticity. The argument is circular.

Furthermore, in Figure 4, the shift in power spectral density (PSD) is not sufficient to conclude elasticity, since there are other materials (e.g., dense colloidal suspensions, laponite) that would be expected to shift the PSD but are not elastic.

Additionally, a theoretical relationship between power spectrum and single-particle displacements is valid for long-time observations. To avoid distortions of the power spectrum (especially, in the low and high frequency parts) due to the finite length of the trajectories, one needs to test if the experimental time frame was long enough.

Paragraph two, subheading “Energy-depleted and acidified cells display increased mechanical stability”: It is stated that an increase in crowding implies an increase in elasticity. This is not necessarily true. What is the evidence for this statement?

In sum, there is insufficient evidence to propose a gel-like state, which implies a physical meshwork. In my opinion, it would be safer to propose a transition into a 'glassy' system (as in Parry et al., 2014). Broadly speaking, the glassy term can applicable to a wide range of out-of-equilibrium soft materials, including gels, foams, interacting or concentrated colloidal suspensions, etc. (Cipelletti and Ramos, Phys Condensed Matter, 2005).

There are also questions about the cell stiffness measurements. If *S. pombe* cells do indeed maintain their rod shape at low pH even after their cell wall has been removed, it would be an absolutely amazing result. This increases the burden of proof. An alternative and trivial explanation for the result is that the cells still have residual cell wall that contributes to the rod shape maintenance and stiffness of the cell. Perhaps enzymatic digestion of the cell wall is incomplete at low pH, which would not be surprising. To me, the maintenance of the rod shape actually argues that wall-free cells were not generated, casting doubt on the claim of increased stiffness. Complete removal of cell wall must be ascertained. Perhaps (round) spheroplasts could first be generated at neutral pH, and then re-shaped (e.g., to rod-like) by an external force (e.g., their pressure-based microfluidic device might do the trick). The authors could then test whether the new (non-round) shape is maintained after a switch to low pH even when the external force is relieved.

Similarly, the authors need to rule out that the increased stiffness observed in energy-depleted/acidified budding yeast cells is not due to the presence of residual cell wall. Stiffness measurements could be performed on spheroplasts created under normal (log phase, neutral pH conditions when the cell wall is known to be completely digested) and then switched to energy-depleted or acidic conditions for stiffness measurements.

Figure 5—figure supplement 2: The statement about the reduction in cell volume is not convincing at this stage. A reduction in cell volume doesn't mean an increase in crowding, unless the reduction in cytoplasmic volume occurred very rapidly before there can be a change in cytoplasmic composition. Under their experimental conditions (30 min), the composition of the cytoplasm can change. Their microfluidic device allows them to measure the cytoplasmic volume immediately following acidification/energy depletion.

Also, what was the osmolality of the external media and buffers used? Can the authors exclude the possibility of an osmotic shock causing the difference in cytoplasmic volume? Also, could they clarify what 'relative cell volume' mean? How many cells were measured?

Figure 5—figure supplement 4: I am not sure what these FRAP curves are showing. Are they for a single cell or for many cells averaged? More importantly, there is no obvious difference in recovery time between the three conditions, contrary to what is stated. Furthermore, the FRAP data should be analyzed, and diffusion coefficients for GFP should be provided. And why was 2M of sorbitol used when the authors suggest that the difference in volume for DNP+pH5.5 cells is equivalent to the sorbitol 0.8 M condition? FRAP measurements under 0.8M sorbitol would be a fairer comparison.

What is the evidence for fractional Brownian motion? There are other types of motion where displacement distributions collapse after re-scaling. It is not clear what fractional Brownian motion brings to the story.

Is the starvation-induced acidification observed regardless of the pH of the external medium? Or is it only when the external medium is acidic? If it is the latter, it should be clearly stated.

Similarly, fission yeast and Dictyostelium are said to undergo cytosolic pH fluctuations in response to energy depletion. Is this true in any medium regardless of its pH or only in acidic media?

How do the MSD curves in Figure 3 compare to Figure 1? If the authors' proposal is correct (reduction in particle dynamics associated with starvation is due to cytosolic acidification), shouldn't we expect the pH 7 condition in Figure 3 to look like the control (untreated/log-phase culture) in Figure 1?

Reviewer #2:

The article is well written and compelling for the most part. The numerous control experiments and the redundant mechanical measurements (microrheology and AFM indentation) support the sol to gel transition hypothesis. There are a few major flaws in the interpretation and presentation of the data that need addressing if the reader is to believe the authors' explanation of pH dependent fluctuations within the cytoplasm. My concerns are below.

1) The use of the term "cytoplamsic freezing" is really questionable. It is common in cell mechanics and active matter to find analogies to inanimate or inactive matter. The "soft glassy rheology" of the cytoskeleton is such an example. This is a good thing to do if there are truly fundamental characteristics shared between the two analogs. In the SGR case, great care was taken to show the analogy. Other examples in cell mechanics have been published to show in detail the analogy between cellular dynamics and some inanimate physical systems. In the submitted manuscript such a level of care is not taken, and the term "freezing" is used without any thought about what are the essential phenomena of freezing, and whether this gelation is anything like true freezing. If the authors believe that what they observe is truly a sol-gel transition associated with assembly of macromolecules in the cytoplasm, they should name the phenomenon accordingly. Their work will be taken more seriously by a broader audience if they let go of this name.

2) A lot of the particle tracking and MSD calculation is performed assuming thermal fluctuations are the driving force. This assumption allows connecting MSDs to material properties. A decent argument is made for this by showing that the depolymerization of actin and microtubules has little effect on their results. I do not know whether this argument is legitimate. Is it really true that no other non-equilibrium or biochemical driving forces can occur in the cytoplasm, other than those driven by the action of ATP on filament bound motors? The work by McIntosh and Schmidt from a few years ago sidestepped this question by comparing active microrheology (manually driven beads with E or B fields) to passive microrheology, and carefully measuring a driving spectrum. The recent paper by Guo and Weitz did something similar. I think this sort of thing would have to be done again here, and the outcome would have to show that the spectrum is indeed thermal, then a thermal analysis could be performed to link MSD to material properties.

Alternatively, the authors could just keep the same analysis and experiments that they have, and tone down their interpretation. I think that most of their story can be told without claiming that they can infer material properties from MSDs. They can just say that the MSDs say things are more "liquid-like" or "solid-like", and then focus on the general phenomena like the dramatic reduction in motion. The supplementary movies show this beautifully. Several years ago many high-profile papers were published that flippantly linked particle MSDs to material properties, which ultimately damaged the reputations of the authors. I think it is imperative that care is taken to prevent the same thing from occurring, for the sake of the authors and the broader research community.

To strengthen their story and focus on a transition in material properties from fluid-like to solid-like, more attention could be paid to the AFM indentation data. They use a Hertz model to fit their data. If they showed the F-d curve on a log-log scale and showed that the indent really scaled like F~d^(3/2)^ (a fit on a lin-lin scale is not sufficient to convince the reader of the right power law), then this would be a first step that they really have a gel. They would also have to show that this could not be attributed to membrane tension, since a balloon filled with water will seem like a solid when indented from the outside. Again, I don't suggest new experiments, but perhaps extracting more compelling information about their current data.

3) The correlation analysis is seriously confusing. At first I thought I was looking at correlation functions, and concluded that the results made no sense. After re-reading the description of their analysis several times, I understood the steps taken by the authors to come up with Figure 4 and the associated supplementary figure. I think the authors need to include a few examples of the displacement autocorrelation functions (as a function of time), and show the process to come up with this correlation map as a function of displacement length.

[Editors’ note: what now follows is the decision letter after the authors submitted for further consideration.]

Thank you for resubmitting your work entitled "A pH-driven transition of the cytoplasm from a fluid- to a solid-like state promotes entry into dormancy" for further consideration at *eLife*. Your revised article has been favorably evaluated by Vivek Malhotra (Senior editor), a Reviewing editor, and two reviewers. Both reviewers agree that the manuscript has been improved but there are some minor issues that need to be addressed before acceptance.

Both reviewers felt that some modifications to the text were warranted as outlined below. One felt that there were some technical concerns and improvements in clarity needed. The other suggests a further consideration be added to the Discussion. We feel that these should be accommodated in the final version.

Reviewer #1:

The revised manuscript is vastly improved, and the additional results on the cell wall-free cells are simply amazing. However, I do have some technical concerns with some of the new analyses. These issues do not affect the major conclusions of the paper, but I believe that the authors should address them before publication.

Figure 4. Why is α measured in the middle of the MSD? The center has fewer statistics, measurements involving the MSD should be made at the beginning where the statistics are the greatest.

Figure 4 and Figure 4—figure supplement 3 and Figure 4—figure supplement 2. I still think that the fractional Brownian motion (fBm) model doesn't bring much to the story (and disrupts the flow), and the evidence is not very strong. Also, a quantitative metric should be used to distinguish Gaussian from non-Gaussian (see equation 1 in Weeks et al., Science, 2000). This can be applied to individual trajectories. The particle size (since it is a mixture) should be considered in the rationale for scaling displacement distributions. What would they see if the distributions were scaled by the relative particle size instead?

It is confusing that Figure 4—figure supplement 5 shows a break in correlation linearity (consistent with caging and cage escape) whereas Figure 4 doesn't show the break (more consistent with purely elastic material). What are the experimental differences and what does it mean to their fBm model?

Figure 4—figure supplement 4. The shape of this correlation plot (a single negative correlation for the first displacement) is precisely what is expected for localization error. They could remove the plot altogether (going back to the support for the fBm model being quite weak). But if they want to show this, the authors should validate that this does not reflect localization error (imperfect localization of a static object will always give this correlation plot). The time scale of the negative correlation should be robust over experiments of different frame rates. If experiments of all frame rates produce anti-correlated displacements over one frame, the result is an artifact.

*Reviewer #2:*

The authors have addressed all of my previous concerns and I recommend publication.

---

## [Author Response]

[Editors’ note: the author responses to the first round of peer review follow.]

We were pleased to hear that Reviewer 1 found our manuscript important: *“This study shows that carbon starvation and acidification of the cytoplasm change the physical properties of the cytoplasm of yeast and Dictyostelium cells. […] The topic is important and has large implications to our understanding of cellular dynamics and dormancy in eukaryotes.”*

Reviewer 2 also found our study compelling and well executed: *“[…]the article is well written and compelling for the most part. The numerous control experiments and the redundant mechanical measurements (microrheology and AFM indentation) support the sol to gel transition hypothesis.”*

Given the overall positive nature of the reviews, we were surprised to hear that our paper was rejected.

The two main reasons for rejecting the paper were the following: *“[…]there are two major problems. First, the authors seemed to propose a 'gel' transition originating from an increase in (presumably cross-linked?) protein filaments in the cytoplasm under starved/acidified conditions, but I find the evidence unconvincing. Second, the AFM and cell shape retention results (that the authors interpret as an increase of stiffness of the cytoplasm) could be due to the presence of residual cell wall. In view of these considerations, we do not feel that the manuscript warrants further consideration unless these concerns can be resolved.”*

As explained below, we can easily resolve these two major concerns.

First, we can provide additional data supporting a gel-like state. We can immediately include additional fluorescence images where assemblies form network-like structures instead of puncta or filaments. We can also add data showing that a large fraction of the cytoplasm becomes insoluble under starvation conditions. However, we agree with the reviewers that the use of the term “gel” is problematic, because it implies a cross-linked network of proteins. We will therefore use more appropriate language to refer to the state of the cytoplasm as “glassy” (as suggested by Reviewer 1) or as “solid-like” (as suggested by Reviewer 2). We think that this resolves the first concern.

The second concern was that there still is a residual cell wall. Reviewer 1 pointed out: *“If S. pombe cells do indeed maintain their rod shape at low pH even after their cell wall has been removed, it would be an absolutely amazing result.”* Convincingly showing that the cell wall is removed is therefore very important, and we should have been more focused on this issue in the paper. Although our previous submission included some supporting data, we now realize that their significance was not emphasized enough. Importantly, other data, which made us confident about complete cell wall removal, was available at the time of submission, but we did not include them in the manuscript. Altogether, we have the following data that supports a complete removal of the cell wall:

1) The submitted manuscript included a movie showing that spheroplasted acidified yeast did not have a cell wall (Video 6). In this movie, a spheroplasted rod-shaped cell can be seen, which was re-exposed to media. After media addition, the cell rapidly rounds up (Figure 8), which indicates that the cell wall had been removed previously.

2) We measured the activity of the enzymes used for spheroplasting and found that they were even more active under low pH than under neutral pH conditions. This is in agreement with information provided by the supplier of the enzymes.

3) We performed calcofluor white staining of spheroplasted cells under starvation and acidification conditions. These data show that the cell wall was completely removed during our spheroplasting procedure (Figure 8).

4) Further evidence for the absence of the cell wall comes from our AFM measurements. While earlier reports had quantified the wild-type yeast stiffness (Young’s modulus) with the cell wall present to be on the order of 1-2 MPa (see Alsteens et al., Nanotechnology, 2008) and with cell wall mutant strains in the range of 0.1–1 MPa (see Dague et al., Yeast, 2010), our results were on the order of 1 kPa, which is 100-1000 times lower. This directly shows that the cell wall had been removed and that we were measuring spheroplasts.

5) Finally, we include a movie (Video 8) showing acidified yeast cells, which were exposed to spheroplasting enzymes at time point zero. Forty minutes later, rod-shaped cells can be seen slipping out of their cell-wall sheath (indicated by the yellow arrows in Figure 8). This directly demonstrates that under the conditions we used, the cell wall was removed.

Taken together, our evidence for cell wall removal is very clear and, in our opinion, leaves no room for alternative interpretations.

Author response image 1.Summary of experiments showing that the cell wall was completely removed in our spheroplasting procedure.(A) A rod-shaped spheroplasted fission yeast cell rounds up when media is added. (B) Acidified control and spheroplasted cells were treated with calcofluor white to stain the cell wall. Note the difference in signal intensity. The graphs on the right show the average calcofluor intensity of ten cells measured through a line scan. The location of the line scan is indicated for two representative cells with a red dashed line. (C) Stills of a time-lapse experiment where acidified rodshaped cells can be seen loosing their cell wall (yellow arrows). Also see corresponding Movie R1**DOI:**
http://dx.doi.org/10.7554/eLife.09347.042

We value the additional comments of the reviewers and appreciate their suggestions, which we think will help us clarify the terminology used in the paper. We agree for example with Reviewer 2 that more care should be taken in the interpretation of the MSD data, in particular with regards to making claims about material properties. We also agree that the term “cytoplasmic freezing” was poorly chosen, and we will use more appropriate language in a revised manuscript. Other minor issues have already been addressed by us experimentally, and can be added to a revised paper.

Like the reviewers, we believe that understanding the physical-chemical mechanisms of dormancy is a very important topic. We hope that, given the overall positive nature of the reviews and our ability to address all major concerns raised in the review process, you reconsider the decision and allow us to resubmit a revised manuscript.

Reviewer #1:

*While the results are potentially exciting, there are major concerns with some interpretations and conclusions.*

We thank the reviewer for asking us to clarify our interpretations and conclusions. In the revised manuscript all these concerns have been addressed through new experiments or textual revisions. We think that these changes have substantially improved the paper, and we are grateful to the reviewer for encouraging us to do this.

*First, the proposed terms 'cytoplasmic freezing' are misleading. If what they observe was analogous to the cytoplasm freezing, all cellular components, including proteins, would reduce their motion as the dynamics of particles of all sizes are affected by temperature. This is not the case here as large particles are affected but not proteins (GFP). The size dependence is inconsistent with a cytoplasmic freezing, and this characterization should be avoided.*

This is a valid point. We used the phrase “cytoplasmic freezing” as an analogy to describe our observations of slower particle movements, but we now have to admit that, from a physical perspective, this term is very misleading for the reasons mentioned by the reviewer. We have therefore changed all appearances of the phrase “cytoplasmic freezing” to the physically more appropriate term “reduced particle mobility”.

*The evidence for a gel transition is not convincing. A gel suggests a meshwork. The authors argue that many proteins assemble into higher order structures under starvation, citing Noree et al. (2010), Narayanaswamy et al. (2009) and their own previous work. The Noree et al. (2010) paper talks about a wide genetic screen identifying metabolic proteins that form 4 distinct filaments, with some being induced by energy depletion, but others being unaffected or even reduced upon energy depletion. The Narayanaswamy et al. (2009) paper and Figure 6 here show mostly puncta forming, but puncta (compact aggregates) do not form gels. I agree that a few filaments form under energy-depleted/acidified conditions. But it is unclear that there are more of them under these conditions than under the normal situation, especially given the loss of cytoskeletal filaments under energy-depletion/acidification? To form a gel and trap probes, the cell would need a pervasive meshwork (cross-linked filaments), not just proteins that form higher-order structures.*

We agree with the reviewer that more direct evidence is required to prove the existence of a gel state, and we have therefore removed all references to a gel. Instead, we now use the less-contentious term solid-like, as also proposed by Reviewer 2. In the Discussion section, we also consider the possibility that the energy-depleted cytoplasm is in a glassy state, as suggested by this reviewer in one of the following comments. Although it would be desirable to make a more detailed statement about the material properties of the cytoplasm, a proper analysis would require substantial experimental effort, and therefore we feel is beyond the scope of the present paper.

*Figure 4: The method used to show that elasticity of the cytoplasm is increased under acidification is based on a model that explicitly assumes elasticity. The argument is circular. Furthermore, in Figure 4, the shift in power spectral density (PSD) is not sufficient to conclude elasticity, since there are other materials (e.g., dense colloidal suspensions, laponite) that would be expected to shift the PSD but are not elastic.*

We agree with the referee that the PSD data alone it not sufficient to claim the changes in elasticity. However, the overall evidence accumulated in the manuscript strongly supports the transition of the cytoplasm to a more solid-like state. We resolve this issue by carefully rephrasing the flow of the text, so that the statements strictly correspond to the data presented to that point. For Figure 4, following the suggestion of Reviewer 2, we focus on the discussion of the displacement data. The PSD plot (previously Figure 4) is now a supplemental figure (Figure 4—figure supplement 1), and we no longer claim to know the exact relation between the PSD and the rheology of the cytoplasm. Instead we mention that in a viscoelastic material driven by thermal fluctuations the similar change in the PSD behavior would correspond to a transition to more solid-like state. We explicitly state that “the thermal nature of driving forces remains an assumption and therefore limits our interpretation of the PSD data”.

*Additionally, a theoretical relationship between power spectrum and single-particle displacements is valid for long-time observations. To avoid distortions of the power spectrum (especially, in the low and high frequency parts) due to the finite length of the trajectories, one needs to test if the experimental time frame was long enough.*

As a control, we randomly reshuffled the pieces of the trajectories to eliminate any temporal correlations. This should correspond to a pure random walk in two dimensions. We show that this leads to Brownian scaling, as expected (Figure 4—figure supplement 1). This indicates that in our data the changing slope in the PSD as a function of frequency is indeed the result of existing correlations in the particle displacements. Additionally, in this response we provide a figure for the PSD obtained for longer trajectories in the control conditions (see Figure 9). The new data consistently follows the same scaling at low frequencies.

Author response image 2.Comparison of the power spectral density of displacements for trajectories of different length (control conditions).The black line corresponds to the data set presented in the paper and the blue line is the new data for 227 trajectories that are 10 min long and acquired with 5 s resolution. As we see from this plot the slope of the PSD for large times (small frequencies) is properly captured by shorter trajectories.**DOI:**
http://dx.doi.org/10.7554/eLife.09347.043

*Paragraph two, subheading “Energy-depleted and acidified cells display increased mechanical stability”: It is stated that an increase in crowding implies an increase in elasticity. This is not necessarily true. What is the evidence for this statement?*

We agree that this statement is not necessarily correct and have removed it.

*In sum, there is insufficient evidence to propose a gel-like state, which implies a physical meshwork. In my opinion, it would be safer to propose a transition into a 'glassy' system (as in Parry et al., 2014). Broadly speaking, the glassy term can applicable to a wide range of out-of-equilibrium soft materials, including gels, foams, interacting or concentrated colloidal suspensions, etc. (Cipelletti and Ramos, Phys Condensed Matter, 2005).*

We agree with the reviewer and appreciate the comment, which we think helped us clarify the terminology in the revised paper. To prevent misunderstanding, we now use the term solid-like to refer to the state of the cytoplasm under energy depletion conditions. We also discuss the possibility of a glass-like state in the Discussion section, and we refer to both papers mentioned by the reviewer.

*There are also questions about the cell stiffness measurements. If S. pombe cells do indeed maintain their rod shape at low pH even after their cell wall has been removed, it would be an absolutely amazing result. This increases the burden of proof. An alternative and trivial explanation for the result is that the cells still have residual cell wall that contributes to the rod shape maintenance and stiffness of the cell. Perhaps enzymatic digestion of the cell wall is incomplete at low pH, which would not be surprising. To me, the maintenance of the rod shape actually argues that wall-free cells were not generated, casting doubt on the claim of increased stiffness. Complete removal of cell wall must be ascertained. Perhaps (round) spheroplasts could first be generated at neutral pH, and then re-shaped (e.g., to rod-like) by an external force (e.g., their pressure-based microfluidic device might do the trick). The authors could then test whether the new (non-round) shape is maintained after a switch to low pH even when the external force is relieved. Similarly, the authors need to rule out that the increased stiffness observed in energy-depleted/acidified budding yeast cells is not due to the presence of residual cell wall. Stiffness measurements could be performed on spheroplasts created under normal (log phase, neutral pH conditions when the cell wall is known to be completely digested) and then switched to energy-depleted or acidic conditions for stiffness measurements.*

We agree with the reviewer that it would be an “absolutely amazing” result, if the cells can keep their shape after cell wall removal. Convincingly showing that the cell wall is removed is therefore paramount, and we should have been more focused on this issue in the paper. Although our previous submission included some supporting data, we now realize that their significance was not emphasized enough. Importantly, other data, which made us confident about complete cell wall removal, was available at the time of submission, but we did not include them in the manuscript. Therefore, in the revised version of the manuscript we now add the following data supporting the conclusion that the cell wall was completely removed in our experiments:

1) The manuscript includes a movie showing that spheroplasted acidified yeast did not have a cell wall (Video 7). In this movie, a spheroplasted rod-shaped cell can be seen, which was re-exposed to media (Figure 5—figure supplement 8). After media addition, the cell rapidly rounds up, which indicates that the cell wall had been removed previously. The cell also starts to divide again, showing that the cells were still alive during the treatment.

2) We measured the activity of the enzymes used for spheroplasting and found that they were similarly active under low and neutral pH conditions. This is in agreement with information provided by the supplier of the enzymes. These data are now included in the manuscript in Figure 5—figure supplement 6.

3) We performed calcofluor white staining of spheroplasted cells under starvation and acidification conditions (see Figure 5—figure supplement 5). These data show that the cell wall was completely removed during our spheroplasting procedure.

4) We performed additional AFM-based indentation measurements on *S. cerevisiae* where the cell wall had not been removed. These measurements resulted in an apparent elastic (Young’s) modulus of 1.3 ± 0.2 MPa (mean ± SEM; N = 69), while the spheroplasted cells had an apparent elastic modulus of around 1 kPa – three orders of magnitude lower (see Figure 10 below). A similar difference in stiffness between spheroplasts and intact cells with a rigid cell wall has previously been found in *E. coli* (Sullivan et al., Ultramicroscopy, 2007). Our values for walled yeast cells are consistent with those reported in the literature for wild-type yeast stiffness (Young’s modulus on the order of 1-2 MPa; see Alsteens et al., Nanotechnology, 2008). These values can drop with cell wall mutant strains by up to a factor of 10 (Young’s modulus in the range of 0.1–1 MPa; see Dague et al., Yeast, 2010), but not by a factor of 1,000. This clearly shows that the cells we analyzed had their cell wall removed. This additional evidence has now been included in Figure 5—figure supplement 1.

5) Finally, we now include a movie showing acidified yeast cells loosing their cell wall (Video 4). The cells in this movie were exposed to spheroplasting enzymes at time point zero. Forty minutes later, rod-shaped cells can be seen slipping out of their cell-wall sheath (indicated by the yellow arrows in Figure 5—figure supplement 8). This directly demonstrates that under the conditions we used, the cell wall was removed.

Author response image 3.AFM based indentation measurements of the apparent elastic modulus of cells with and without cell wall.In both cases, the cells were indented up to a maximum force of 2 nN and the resulting force-distance curves were analyzed with the Hertz model as described in the material and Methods section. Note the logarithmic scale of the y-axis.**DOI:**
http://dx.doi.org/10.7554/eLife.09347.044

Taken together, our evidence for cell wall removal is very clear. Therefore, in our opinion, all results obtained with yeast spheroplasting experiments can only be explained by assuming a transition of the cytoplasm from a fluid-like to a more solid-like state.

*Figure 5—figure supplement 2: The statement about the reduction in cell volume is not convincing at this stage. A reduction in cell volume doesn't mean an increase in crowding, unless the reduction in cytoplasmic volume occurred very rapidly before there can be a change in cytoplasmic composition. Under their experimental conditions (30 min), the composition of the cytoplasm can change. Their microfluidic device allows them to measure the cytoplasmic volume immediately following acidification/energy depletion.*

The reviewer is correct to point out that a change in cell volume does not necessarily lead to an increase in molecular crowding. We also agree that the time at which the cell volume measurement is performed is very important. For this reason, we did further experiments with our microfluidic device (RT-DC) to test the time dependence of the cell volume reduction. We found a linear and rather slow reduction in cell volume over a time span of 30 min in response to low pH conditions, as shown below in Figure 11. We agree with the reviewer that this slow change in cell volume would allow for changes in the composition of the cytoplasm and does therefore not necessarily imply increased molecular crowding. We have rephrased the paragraph and now make clear that a volume decrease can, but does not have to, lead to an increase in crowding.

Author response image 4.The cell size of *S. cerevisiae* cells in response to low pH adjustment was measured over time with an RT-DC setup.Cells were loaded into the microfluidic chip of the RT-DC device immediately after exposure to phosphate buffer of pH 6 containing 2 mM DNP and 2% glucose. The median diameter of more than 530 cells was measured for each time-point. We observed a small but significant decrease in cell size over a time span of 30 min. A two sided t-test shows that the slope differs significantly from zero.**DOI:**
http://dx.doi.org/10.7554/eLife.09347.045

*Also, what was the osmolality of the external media and buffers used? Can the authors exclude the possibility of an osmotic shock causing the difference in cytoplasmic volume?*

We measured the osmolality of the external media and buffers to exclude the possibility that the observed cell volume changes are caused by an osmotic shock. The results of these measurements are now included as a supplementary table (Table S3). We did not observe differences in the osmolality that would explain the observed differences in cytoplasmic volume.

*Also, could they clarify what 'relative cell volume' mean? How many cells were measured?*

The term “relative cell volume” was used to indicate that the volume of control cells (cells in SD medium containing all nutrients) was normalized to 100%. The volume of cells exposed to low pH or sorbitol is then expressed relative to the volume of these control cells. For each condition the volume of more than 160 cells was measured. This is now mentioned in the figure legend.

*Figure 5—figure supplement 4: I am not sure what these FRAP curves are showing. Are they for a single cell or for many cells averaged? More importantly, there is no obvious difference in recovery time between the three conditions, contrary to what is stated. Furthermore, the FRAP data should be analyzed, and diffusion coefficients for GFP should be provided. And why was 2M of sorbitol used when the authors suggest that the difference in volume for DNP+pH5.5 cells is equivalent to the sorbitol 0.8 M condition? FRAP measurements under 0.8M sorbitol would be a fairer comparison.*

We agree and have improved and repeated these experiments with a mCherry-GFP fusion protein, and added a description in the Materials and methods section and in the figure legend. We also included measurements for different sorbitol concentrations (0.8 M, 1 M, 1.5 M and 2 M sorbitol) as suggested by the reviewer. The results of these new experiments are now included as Figure 5—figure supplement 5 and show – as correctly stated by the reviewer – that there is no obvious difference in recovery time between control and low pH adjusted cells. Only in cells subjected to strong osmotic compression (1.5 M sorbitol) could we detect a significant difference in the recovery time. The fact that mCherry-GFP diffusion is not impaired under low pH conditions indicates the presence of a fluid phase that allows unimpaired diffusion of mCherry-GFP. This argues against osmotic compression as the mechanism and is in agreement with a previous paper on the glassy nature of the bacterial cytoplasm (Parry et al., 2014). This is now stated more clearly in the text, and further considered in the Discussion section.

*What is the evidence for fractional Brownian motion? There are other types of motion where displacement distributions collapse after re-scaling. It is not clear what fractional Brownian motion brings to the story.*

Several complementing pieces of evidence point to fractional Brownian motion (fBm) as a relevant model. Rescaling of distributions for different lag times is a necessary step to demonstrate that the displacements can be characterized by a self-similar distribution. The corresponding time scaling is sub-diffusive, which indeed can be accommodated by several common statistical models of sub-diffusion. As we now show in the revised version of the manuscript, the probability distribution function (PDF) of displacements obtained for individual trajectories is Gaussian (see Figure 4—figure supplement 2). The PDF of displacements of all trajectories can be collapsed onto a single Gaussian master curve via an additional rescaling of displacements by the corresponding generalized diffusivities of individual trajectories (see Figure 4—figure supplement 3). Finally, from our data we see that displacements are negatively correlated. These key features (sub-diffusive scaling, Gaussian profile of PDF, no ageing, and negative correlations) narrow down the choice of known (common) models to the model of fBm.

Analysis of the experimental data through the prism of fBm brings several important consequences. We can use the model to interpret the difference between the experimental conditions in a very low-dimensional parameter space: fBm is fully characterized by its scaling exponent and the generalized diffusion constant. Deviations of the PDF of the ensemble of trajectories from a Gaussian distribution give access to the variability of the generalized diffusivities and therefore to the heterogeneity of the particles’ environment. Furthermore, the strength of negative correlations makes a link to the changing elastic properties of the cytoplasm.

*Is the starvation-induced acidification observed regardless of the pH of the external medium? Or is it only when the external medium is acidic? If it is the latter, it should be clearly stated.*

Cytosolic acidification is only observed when the external medium is acidic, which is now clearly stated in the text and in the figure legends.

*Similarly, fission yeast and Dictyostelium are said to undergo cytosolic pH fluctuations in response to energy depletion. Is this true in any medium regardless of its pH or only in acidic media?*

We believe that *Dictyostelium* and fission yeast – like budding yeast – only undergo cytosolic pH fluctuations when the outside medium or buffer is acidic. For these two organisms, we also used DNP to adjust the cytosolic pH, and we found that the particle movement is not impaired in neutral media (see Figure 3). We therefore assume that energy depletion causes a passive adjustment of the cytosolic pH to the pH of the external medium, as also observed by us in budding yeast. This is in agreement with previous observations that *Dictyostelium* and *S. pombe* undergo cytosolic pH fluctuations of a similar magnitude (Gross et al., 1983; Karagiannis and Young, 2001). Furthermore, these organisms preferably live in acidic environments and presumably spend a lot of energy to maintain their cytosolic pH via ATP-dependent proton pumps. Unfortunately, our molecular understanding of these organisms is very limited, and further work is required to learn more about the link between energy depletion and cytosolic pH. However, we would like to point out that acidification through outside protons may not be the only mechanism of acidifying the cytosol. In fact, a previous paper, which reported intracellular pH fluctuations in dormant marine brine shrimp, implicated intracellular stores as proton source (Covi et al., 2005). We now clarify the role of the outside pH in the text, and we added additional explanations and citations to the Discussion section to discuss the broader implications.

*How do the MSD curves in Figure 3 compare to Figure 1? If the authors' proposal is correct (reduction in particle dynamics associated with starvation is due to cytosolic acidification), shouldn't we expect the pH 7 condition in Figure 3 to look like the control (untreated/log-phase culture) in Figure 1?*

The experiments in Figure 1 were performed with cells growing in medium, whereas the control MSDs in Figure 3 were performed with cells treated with DNP at neutral pH (Figure 3) or with 1 mM sorbic acid (Figure 3). We consistently saw a difference between the first condition and the latter two conditions, and we can only speculate about the causes. First, both treatments slightly change the cytosolic pH and may also change the concentrations of other ions, which could already have an effect on particle mobility. Second, DNP equilibrates proton gradients over all membranes in the cell. It may therefore perturb the proton gradient over the inner mitochondrial membrane (this is what DNP is normally used for), thus inhibiting mitochondrial ATP production. Thus, cells treated with DNP or low amounts of sorbic acid could already experience some form of energy depletion, which could lead to the partial reduction in MSD that we observe. We clearly state this in the Results section.

*Reviewer #2:*

*1) The use of the term "cytoplamsic freezing" is really questionable. It is common in cell mechanics and active matter to find analogies to inanimate or inactive matter. The "soft glassy rheology" of the cytoskeleton is such an example. This is a good thing to do if there are truly fundamental characteristics shared between the two analogs. In the SGR case, great care was taken to show the analogy. Other examples in cell mechanics have been published to show in detail the analogy between cellular dynamics and some inanimate physical systems. In the submitted manuscript such a level of care is not taken, and the term "freezing" is used without any thought about what are the essential phenomena of freezing, and whether this gelation is anything like true freezing. If the authors believe that what they observe is truly a sol-gel transition associated with assembly of macromolecules in the cytoplasm, they should name the phenomenon accordingly. Their work will be taken more seriously by a broader audience if they let go of this name.*

We completely agree and have corrected this. We had used the phrase “cytoplasmic freezing” as a mere analogy to describe our observations of slower particle movements, but did not mean to imply an actual freezing in the real physical meaning of the term. We have therefore changed all appearances of the misleading phrase “cytoplasmic freezing” to the more appropriate term “reduced particle mobility”.

*2) A lot of the particle tracking and MSD calculation is performed assuming thermal fluctuations are the driving force. This assumption allows connecting MSDs to material properties. A decent argument is made for this by showing that the depolymerization of actin and microtubules has little effect on their results. I do not know whether this argument is legitimate. Is it really true that no other non-equilibrium or biochemical driving forces can occur in the cytoplasm, other than those driven by the action of ATP on filament bound motors? The work by McIntosh and Schmidt from a few years ago sidestepped this question by comparing active microrheology (manually driven beads with E or B fields) to passive microrheology, and carefully measuring a driving spectrum. The recent paper by Guo and Weitz did something similar. I think this sort of thing would have to be done again here, and the outcome would have to show that the spectrum is indeed thermal, then a thermal analysis could be performed to link MSD to material properties.*

We fully agree with the referee that in such an inherently out-of-equilibrium and active medium as the cytoplasm, a reliable measurement of material properties can be achieved only by a combination of active and passive microrheology, which we now also state in the text. At the moment, however, active microrheology in yeast cells is extremely challenging because of their small size and stiff cell wall and cannot be achieved in a reasonable time frame. Therefore, we followed the alternative suggestion of the referee below.

*Alternatively, the authors could just keep the same analysis and experiments that they have, and tone down their interpretation. I think that most of their story can be told without claiming that they can infer material properties from MSDs. They can just say that the MSDs say things are more "liquid-like" or "solid-like", and then focus on the general phenomena like the dramatic reduction in motion. The supplementary movies show this beautifully. Several years ago many high-profile papers were published that flippantly linked particle MSDs to material properties, which ultimately damaged the reputations of the authors. I think it is imperative that care is taken to prevent the same thing from occurring, for the sake of the authors and the broader research community.*

We thank the reviewer for proposing an alternative way of addressing this issue. We completely rephrased the part of the text when discussing the connection of the particle displacements and material properties. We stress that both active and passive microrheology measurements would be necessary to recover the material properties of the cytoplasm. The assumption of thermal fluctuations is indeed at best approximate for cells under normal conditions. However, it becomes more reasonable for acidified and energy depleted cells, where we expect most of the chemical reactions to shut down and proteins to form assemblies. Still, we now use the example of the near-thermal fluctuations only as a way to illustrate the relative changes in apparent properties of the cytoplasm, hinting at the build-up of elastic response for acidified and energy depleted cells. We hope that this is now adequately worded in the text.

*To strengthen their story and focus on a transition in material properties from fluid-like to solid-like, more attention could be paid to the AFM indentation data. They use a Hertz model to fit their data. If they showed the F-d curve on a log-log scale and showed that the indent really scaled like F~d^(3/2)^ (a fit on a lin-lin scale is not sufficient to convince the reader of the right power law), then this would be a first step that they really have a gel. They would also have to show that this could not be attributed to membrane tension, since a balloon filled with water will seem like a solid when indented from the outside. Again, I don't suggest new experiments, but perhaps extracting more compelling information about their current data.*

We appreciate this suggestion to glean more information from the existing AFM data in order to support the fluid-to-solid transition in the yeast cytoplasm under acidification. AFM-based indentation experiments on biological cells have most commonly been used to extract an apparent elastic modulus using the Hertz model – and our experiments were set up to do just that. The 2.5-fold increase in apparent elastic modulus (extracted from the fit to the Hertz model) from *E* = 636 ± 16 Pa at pH 7.4 to E = 1459 ± 59 Pa at pH 6.0 is certainly consistent with an increased solidification. However, as the reviewer rightly points out, this does not prove a transition from fluid- to solid-like as the Hertz model assumes an elastic solid. The fact that we measure an increased apparent elastic modulus could also be due to an increase in viscosity, and a large viscous resistance to deformation, which we wrongly assign to an increased elastic resistance to deformation.

In order to address this ambiguity, we have now further analyzed the existing force-distance curves during the indentation-retraction cycle and extracted the viscosity of the cells following the approach published by the group of Manfred Radmacher (Rebelo et al., 2013; for details see Methods section). We find that the viscosity decreases from *η*≈ 90 Pa s to *η*≈ 70 Pa s (see Figure 12). There is not only no 2.5-fold increase in viscosity, which could otherwise account for the increase in apparent elastic modulus previously reported, the viscosity is actually decreasing. This also means that the actual elastic modulus increases by more than a factor of 2.5 from pH 7.4 to pH 6.0.

Author response image 5.Viscosity of *S. cerevisiae* spheroplasts at pH 7.4 and pH 6.0 measured by AFM nano-indentation experiments and analyzed according to Rebelo et al.The viscosity reduces from *η*= 90 ± 16 Pa s (mean ± SEM; *N* = 31; median is 81 Pa s) at pH 7.4 to *η* = 70 ± 14 Pa s (*N* = 23; median is 59 Pa s) at pH 6.0.**DOI:**
http://dx.doi.org/10.7554/eLife.09347.046

The fluid-to-solid transition can also be expressed in terms of a phase angle. Since the viscosity is proportional to the loss modulus (as a crude estimate *E”* = *ηω*, with *ω*≈ 2*π* / 2 sec in our case), we can estimate the phase angle to change from about 25° to less than 10°, indicating a solidification. We are hesitant to report these phase angle values in the manuscript, as they contain very crude assumptions. However, the reduction in viscosity, together with the increase in elasticity necessitates a reduction in phase angle and clearly demonstrates that the cells transition from a compliant, more viscous material to a stiffer, more elastic material at lower pH. We have added this new additional analysis to the main manuscript, which now further strengthens our main message.

We also considered the suggestion made by the reviewer of plotting the *F-d* data in a log-log plot to test for 3/2 slope, which holds for a conical indenter. However, since we used a spherical indenter, which does not have a simple power-law dependence, this analysis did not yield any useful information.

We agree that some of the elastic resistance to deformation is likely contributed by membrane tension, as the reviewer stipulates. However, it is unlikely that all of the increase in stiffness comes from an increase in membrane tension. In fact, other evidence presented in the manuscript suggests a strong contribution from intracellular elastic elements, as evident by the strong increase in elasticity, which we extract from the SPT experiments inside the cytoplasm, and the lack of rounding-up of yeast cells at pH 6.0, when the cell wall is removed (see Figure 5, Video 4). In fact, higher membrane tension at low pH should lead to faster rounding up, and not a lack of rounding up. Furthermore, we observed that the cell volume decreases under low pH conditions, which should, if anything, lead to more available membrane and lower membrane tension in acidified yeast. The emerging picture, consistent with all available evidence, is thus more of a balloon filled with a fluid at pH 7.4 and an elastic bulk material at pH 6.0 with membrane tension changes playing a minor role.

*3) The correlation analysis is seriously confusing. At first I thought I was looking at correlation functions, and concluded that the results made no sense. After re-reading the description of their analysis several times, I understood the steps taken by the authors to come up with Figure 4 and the associated supplementary figure. I think the authors need to include a few examples of the displacement autocorrelation functions (as a function of time), and show the process to come up with this correlation map as a function of displacement length.*

We thank the reviewer for this comment. We reworked the corresponding part of the text discussing negative correlations. First the standard displacement correlation function as a function of the lag time is calculated, and it shows negative correlations (Figure 4—figure supplement 4). Next, we look at the strength of correlations between two consecutive steps. For that we calculate the average projection of the second step on the direction of the initial step and plot it as a function of the initial displacement length (Figure 4 and Figure 4—figure supplement 5). This allows us to test if a large initial fluctuation in a particle’s position is followed by a proportionally larger displacement in the opposing direction at the next instant of time. The (negative) slope of the linear dependence connecting the strength of negative correlation to the length of the initial step varies from 0 (viscous material) to -1/2 (elastic material). For our data we can see that the negative slopes larger in magnitude correspond to energy-depleted or acidified cells. We also make sure that this result does not change, if we vary the lag time for which the displacements are calculated.

[Editors’ note: the author responses to the re-review follow.]

Reviewer #1:

*Figure 4. Why is α measured in the middle of the MSD? The center has fewer statistics, measurements involving the MSD should be made at the beginning where the statistics are the greatest.*

Reviewer 1 correctly points out that the statistics are greatest at the beginning of the MSD. However, we are interested in the longer time asymptotic behavior of the system, or a time span during which the displacements of the particles have the same scaling properties. Therefore measuring the slope of the MSD in the middle region (0.2 to 2 s) is a compromise between the long times and still sufficient statistics. Moreover, expanding the fitting region (0.05 to 5 s) does not significantly change the resulting exponents: cells at pH 6.0: 0.77 (previously 0.73), cells at pH 7.4: 0.81 (previously 0.84), energy-depleted cells: 0.63 (previously 0.64), log phase cells: 0.87 (previously 0.88).

*Figure 4 and Figure 4—figure supplement 3 and Figure 4—figure supplement 2. I still think that the fractional Brownian motion (fBm) model doesn't bring much to the story (and disrupts the flow), and the evidence is not very strong.*

We agree that this very theoretical part affects the overall flow of the manuscript, but we think that it is important to have a coherent view of the experimental data, a consistent comparison, and a characterization of the cytoplasmic changes with a small number of parameters. Moreover, the additional tests suggested by Reviewer 1 further support the consistency of the fBm model with our tracking data (see below). We therefore think the model should be discussed in the manuscript.

*Also, a quantitative metric should be used to distinguish Gaussian from non-Gaussian (see equation 1 in Weeks et al., Science, 2000). This can be applied to individual trajectories.*

The suggested formula defines the non-Gaussian parameter as (*kurtosis*/3-1), where *kurtosis* is the ratio of the fourth and the square of the second moments of distribution. For a Gaussian distribution the value of this parameter is zero. We followed the advice of the reviewer and calculated this parameter for each individual trajectory for each experimental condition. The corresponding scatter plot (now included in the inset in Figure 4—figure supplement 2) indeed shows that they are all close to zero. The corresponding text is added to the Materials and methods section. Note that the non-Gaussian parameter being equal to zero is a necessary but not sufficient condition for a Gaussian distribution. Therefore the CDFs shown in the same figure provide further evidence for the Gaussian behavior of individual trajectories.

*The particle size (since it is a mixture) should be considered in the rationale for scaling displacement distributions. What would they see if the distributions were scaled by the relative particle size instead?*

This point was already addressed in the manuscript. We rescale the displacements of particles by their generalized diffusivities, which contain effects of both the particle size and the heterogeneous environment. However, as shown in Figure 1—figure supplement 3, even for particles of similar sizes, the diffusivities can vary by two orders of magnitude. Therefore, just rescaling by the particle size does not collapse the data.

*It is confusing that Figure 4—figure supplement 5 shows a break in correlation linearity (consistent with caging and cage escape) whereas Figure 4 doesn't show the break (more consistent with purely elastic material). What are the experimental differences and what does it mean to their fBm model?*

The experimental conditions in in Figure 4 and Figure 4—figure supplement 5 are indeed different: (WT and energy-depleted) and (pH 7.4 and 6 induced by DNP), respectively. However, the apparent deviation from a linear behavior is in the regime of large displacements with very limited statistics. We would therefore like to refrain from over-interpreting this part of the data.

*Figure 4—figure supplement 4. The shape of this correlation plot (a single negative correlation for the first displacement) is precisely what is expected for localization error. They could remove the plot altogether (going back to the support for the fBm model being quite weak). But if they want to show this, the authors should validate that this does not reflect localization error (imperfect localization of a static object will always give this correlation plot). The time scale of the negative correlation should be robust over experiments of different frame rates. If experiments of all frame rates produce anti-correlated displacements over one frame, the result is an artifact.*

As discussed in (Weber et al., 2012) one of the ways to test for localization error is to plot the normalized velocity autocorrelation functions for increasing time steps (which are used to determine velocity from the displacement data). For the localization error, the negative correlations will approach zero with increasing lag time. We provide such a plot below and show that the negative correlations do not disappear even for very long lag times. In fact the existing plots in the manuscript already contain similar information: we see that a negative correlation is not just constant but increases as a function of the previous step length, and it remains unchanged for the different lag time used to define the displacements (see Figure 4 for example). We put a corresponding sentence in the Materials and methods subsection discussing displacement correlations.

Author response image 6.Normalized velocity autocorrelation function plotted for different lag times τ from 30 ms to 500 ms, which are used to define the velocity from the displacement data (the lag time change is color-coded from blue to red in steps of 10 ms).Non-vanishing negative correlations indicate that they are not caused by a localization error. Left and right panels show log phase and energy depleted cells data respectively.**DOI:**
http://dx.doi.org/10.7554/eLife.09347.047